# Molecular signatures of cortical expansion in the human foetal brain

G. Ball [1,2] ✉, S. Oldham [1], V. Kyriakopoulou [3,4], L. Z. J. Williams[3,4], V. Karolis [3,5], A. Price[3,4], J. Hutter[3,4], M. L. Seal[1,2], A. Alexander-Bloch [6,7,8,9], J. V. Hajnal [3,4], A. D. Edwards [3,4], E. C. Robinson [3,4] & J. Seidlitz [6,7,8,9]

The third trimester of human gestation is characterised by rapid increases in brain volume and cortical surface area. Recent studies have revealed a remarkable molecular diversity across the prenatal cortex but little is known about how this diversity translates into the differential rates of cortical expansion observed during gestation. We present a digital resource, μBrain, to facilitate knowledge translation between molecular and anatomical descriptions of the prenatal brain. Using μBrain, we evaluate the molecular signatures of preferentially-expanded cortical regions, quantified in utero using magnetic resonance imaging. Our findings demonstrate a spatial coupling between areal differences in the timing of neurogenesis and rates of neocortical expansion during gestation. We identify genes, upregulated from mid-gestation, that are highly expressed in rapidly expanding neocortex and implicated in genetic disorders with cognitive sequelae. The μBrain atlas provides a tool to comprehensively map early brain development across domains, model systems and resolution scales.

The human cortex is a tapestry of specialised cortical areas supporting diverse and complex behaviours, each identifiable on the basis of distinct patterns of cytoarchitecture, chemo-architecture, and axonal connectivity[1–5]. During gestation, waves of neurons are generated from progenitor cells lining the cerebral ventricles and migrate outwards along supporting radial glia to form the layers of the cortex[6–8]. Prior to the ingress of extrinsic connections via the thalamus[9], the progressive differentiation of cortical areas is orchestrated by transcription factors expressed along concentration gradients and translated from the ventricular zone (VZ) to secondary progenitors of the subventricular zone (SVZ), then onto neurons in the cortical plate (CP), forming functional territories[6,10–14]. This process follows a precise spatio-temporal schema[6,14–17], the traces of which extend far beyond the nascent stages of neurogenesis and are echoed in patterns of cytoarchitecture, axonal connectivity and function[18–24].

Focused on uncovering the mechanisms that govern areal differentiation, studies have begun to catalogue the cellular diversity of the developing human cortex and genes that encode it, with increasing granularity and scale[10,25–28]. Regional specialisation of cell types has been observed from early in gestation, with a diversity of cortical gene transcription most evident in mid- to late-gestation but persisting into adulthood and aligning with structural and functional organisation of the brain[21,29–33].

The third trimester of human gestation is characterised by rapid and sustained increases in brain volume and cortical surface area[6,34,35]. Differential rates of areal expansion during human development

[1]Developmental Imaging, Murdoch Children's Research Institute, Melbourne, Australia. [2]Department of Paediatrics, University of Melbourne, Melbourne, Australia. [3]Centre for the Developing Brain, King's College London, London, UK. [4]School of Biomedical Engineering & Imaging Science, King's College London, London, UK. [5]Wellcome Centre for Integrative Neuroimaging, FMRIB, Nuffield Department of Clinical Neurosciences, University of Oxford, Oxford, UK. [6]Department of Child and Adolescent Psychiatry and Behavioral Sciences, The Children's Hospital of Philadelphia, Philadelphia, PA, USA. [7]Department of Psychiatry, University of Pennsylvania, Philadelphia, PA, USA. [8]Lifespan Brain Institute, The Children's Hospital of Philadelphia and Penn Medicine, Philadelphia, PA, USA. [9]Institute of Translational Medicine and Therapeutics, University of Pennsylvania, Philadelphia, PA, USA. ✉e-mail: gareth.ball@mcri.edu.au

mirror evolutionary trends in cortical scaling and function[36–41] with preferential expansion in areas vulnerable to disruption in neurodevelopmental[42], neurological[43], genetic[44] and psychiatric[45] disorders. Juxtaposed hypotheses implicate either the production of glia[46,47] or neurons[48–51] from specialised progenitor populations of the outer SVZ, in the expansion of the primate cortex. Thus, the distribution of distinct cell populations across the developing cortex may mediate areal differences in expansion and vulnerability to insult[14,30], but we currently do not have a clear understanding of how this molecular diversity is translated into cortical organisation in humans in vivo.

## Results

### µBrain: A three-dimensional microscale atlas of the foetal brain

To bridge this gap, we sought to construct a 3D digital atlas of the developing brain at a micrometre scale using a public resource of 81 serial histological sections of a prenatal human brain at 21 post-conceptional weeks (PCW)[29,52]. Source data included serial coronal sections (20µm thickness) obtained from the right hemisphere of a single prenatal brain specimen (21 PCW; female), Nissl-stained, imaged at 1-micron resolution and labelled with detailed anatomical annotations, alongside interleaved coronal sections stained with in situ hybridisation (ISH) of n = 41 developmental gene markers, as reported by Ding et al.[52] (Supplementary Fig. S1 and Supplementary Data S1–S3; see "Methods"). In this and 3 other specimens (15, 16 and 21 PCW, 2 female), anatomical annotations had been used to guide a series of laser microdissections (LMD) across multiple cortical areas and layers of the cortical anlage (e.g., cortical plate, subplate, intermediate zone, ventricular zone; Supplemental Data S4) in the left hemisphere to measure regional gene expression via RNA microarrays[29]. Nissl- and ISH-stained sections with corresponding anatomical labels and LMD arrays were made available as part of the BrainSpan Developing Brain Atlas [https://atlas.brain-map.org/atlas?atlas=3].

Artefacts due to tissue preparation, sectioning and staining procedures (including tearing and folding of sections) are common in histological data and can present difficulties for downstream processing pipelines[53–56]. To correct for tissue artefacts present in the histological data, we designed an automated detect-and-repair pipeline for Nissl-stained sections based on pix2pix, a Generative Adversarial Network (GAN)[57,58] (Fig. 1a–d; see "Methods"). Using 256 × 256 pixel image patches (n = 1000) drawn from 73/81 labelled histological sections (n = 8 reserved for model testing) with paired anatomical labels, we trained a GAN model to produce Nissl-contrast images conditioned on a set of 20 tissue labels (Fig. 1a and Supplementary Data S2). After training, the model was able to produce realistic, Nissl-stained image patches matched on colour hue and saturation to the original data using tissue annotations alone (Fig. 1c). Model performance was robust to different parameter settings and model architectures (Supplementary Fig. S2). Using the trained model, we generated synthetic Nissl-contrast image predictions from anatomical annotations of each section and identified artefacts in the histological data based on deviations in pixel hue and saturation from the model prediction. Outlier pixels were replaced with model predictions using Poisson image editing[59] (Fig. 1d), resulting in n = 79 (2 excluded due to extensive missing tissue) complete histological sections (Supplementary Fig. S1 and Supplementary Data S1).

Histological atlases of the cerebral cortex[2,5] have proven invaluable for understanding human brain organisation but are limited by the loss of spatial information inherent to 2D representations of 3D structures. Reconstructions of 3D brain volumes from serial tissue sections of post-mortem tissue allow the examination of intact brain anatomy at a scale inaccessible to current neuroimaging technologies[60]. We combined repaired tissue sections into a 3D volume of the right hemisphere using iterative affine image registration constrained by a tissue shape reference derived from foetal

MRI (Supplementary Fig. S3; see "Methods")[61], followed by nonlinear alignment to account for warping between adjacent sections. Using the aligned data, we generated a 3D volume resampled to voxel resolution 150 × 150 × 150 µm with dimension 189 × 424 × 483 voxels (28.35 × 63.60 × 72.45 mm) (µBrain; see "Methods"; Fig. 1e and Supplementary Fig. S4a, b). Following reconstruction, we benchmarked the size of the reconstructed µBrain volume against standard foetal growth metrics for a 23-week (gestational age; GA, equivalent to 21 PCW) foetus (µBrain length = 62.7 mm, 23 weeks GA occipital-frontal diameter median [5th, 95th centile] = 73.3 mm [68.2, 78.5])[62], and compared tissue volume estimates based on reconstructed anatomical labels (parenchymal volume = 25.8 ml, right hemisphere) to previously reported 3D MRI-derived foetal brain volumes (supratentorial volume [both hemispheres] at 23 weeks GA = 60.26 ml)[63]. Adapting protocols from neuroimaging analysis, we extracted the inner and outer surfaces of the cortical plate and projected a set of 29 cortical area labels derived from the histological tissue annotations (Supplementary Data S2) onto the surface vertices (Fig. 1f). This resulted in the µBrain cortical atlas, a parcellation of the developing human cortex defined according to the hierarchical ontology of the reference annotations and matched to corresponding LMD microarray data (Supplementary Data S2, S4 and Supplementary Fig. S4c, d).

In addition to the whole brain volume and cortical atlas, we created partial 3D reconstructions of ISH staining for 41 genes (see "Methods"; Fig. 1g). Based on an average 41 tissue sections per gene (Supplemental Data S3), semi-quantitative maps of gene expression revealed the tissue- and region-specific distributions of several genes, including caudal enrichment of the transcription factor EOMES in the subventricular zone[11], and markers of neuronal migration (DCX[64]) and synaptic transmission (GRIK2[65]), in the cortical plate (Fig. 1g and Supplementary Fig. S5).

Existing histological brain atlases, including those of the adult human[60,66,67], mouse[68,69], and macaque[70] brains, facilitate integration with other data modalities, including neuroimaging, and are amenable to advanced computational image analysis methods to extract quantitative measures of neuroanatomy across multiple scales[71,72]. Building upon existing resources[29,52], we have created the µBrain atlas (Fig. 1 and Supplementary Fig. S4), a freely available 3D volumetric model of the 21 PCW foetal brain at 150µm resolution, accompanied by a set of n = 20 cerebral tissue labels (Supplementary Fig. S4a, b); surface models of the cortical plate surface and cortical plate/subplate interface with n = 29 cortical area labels (Supplementary Fig. S4c) and n = 41 partial reconstructions of ISH expression data (Supplementary Fig. S5). Cortical areas are matched to normalised gene expression data from corresponding LMD microarrays (Supplementary Data S4 and Supplementary Fig. S4d) across multiple tissue zones in three additional prenatal specimens (total n = 4), providing a 3D anatomical coordinate space to facilitate integrated imaging-transcriptomic analyses of the developing brain. Below, we use the µBrain atlas to evaluate the molecular and cellular correlates of cortical expansion in the third trimester of human gestation.

### Tissue- and region-specific gene expression in the mid-gestation brain

We sought to characterise patterns of gene expression in the mid-gestation brain and identify developmental and region-specific genes with putative roles in cortical expansion. To do so, we used publicly available microarray data from four prenatal brain specimens aged 15 to 21 PCW[29]. Microarray probe annotations were updated, and tissue samples matched to the µBrain atlas (Supplementary Data S4), yielding expression data of 8771 genes sampled from between 18 and 27 brain regions and across 5 tissue zones for each specimen (see "Methods"; Supplementary Fig. S4d). Applying PCA to gene expression profiles, we found that tissue samples were primarily separated according to

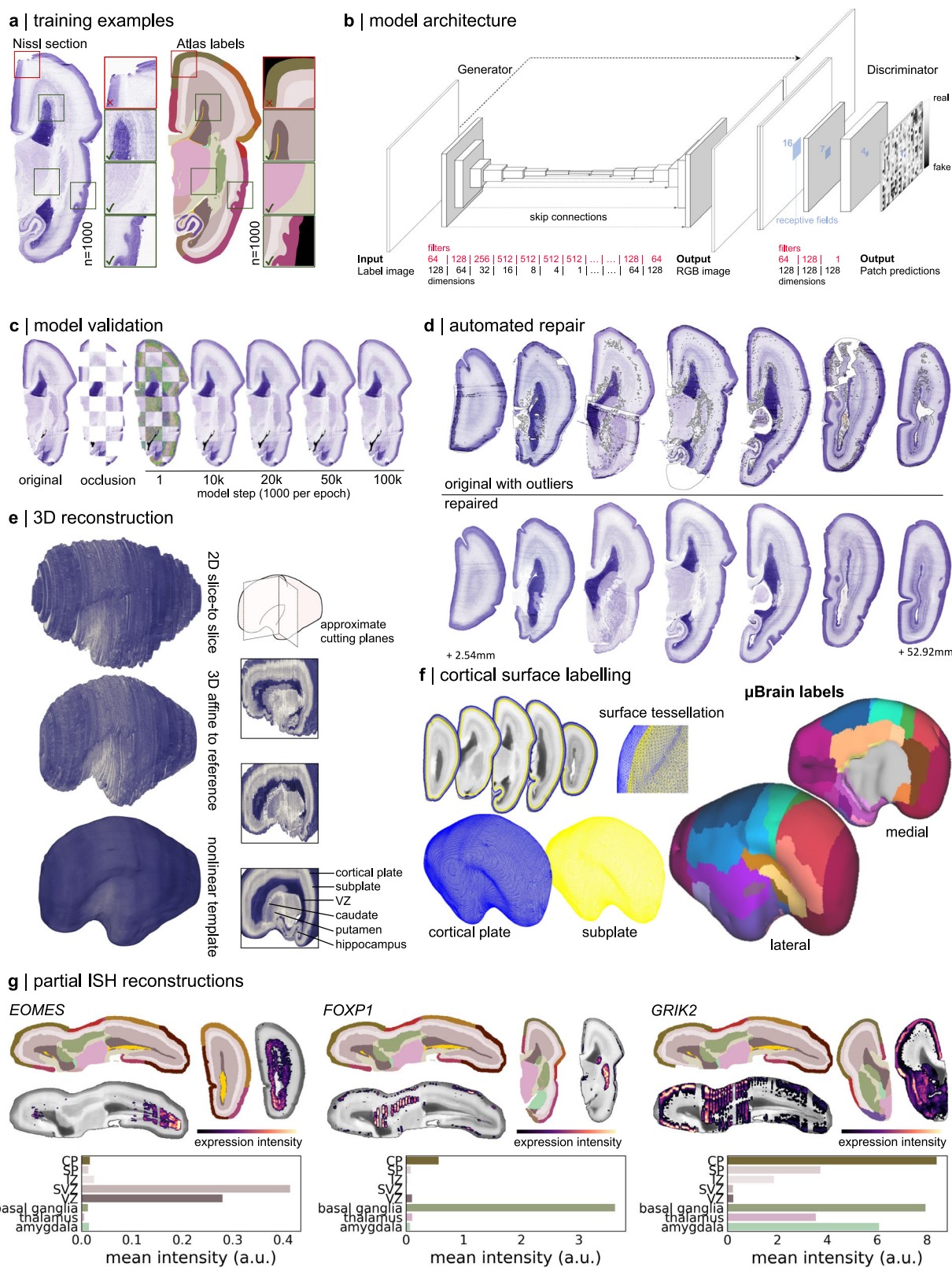

**a | training examples**

**b | model architecture**

**c | model validation**

**d | automated repair**

**e | 3D reconstruction**

**f | cortical surface labelling**

**g | partial ISH reconstructions**

location in mitotic (VZ, SVZ) or post-mitotic tissue zones, rather than across regions (Fig. 2a) – a pattern that was replicated across all specimens when analysed separately (Supplementary Fig. S6). Focusing on expression profiles within each tissue zone, samples clustered according to maturity (Fig. 2a and Supplementary Fig. S7) with developmental changes in gene expression most similar across adjacent mitotic (SVZ and VZ, $r = 0.43$) and post-mitotic zones (CP and SP,

$r = 0.67$, Supplementary Fig. S8). Examining changes over time, we observed increased expression of genes enriched in post-mitotic excitatory neurons and interneurons (e.g., *GRIK1-3*; *GLRA2*; *SCN3B*) between 15 and 21 PCW in the VZ, in line with evidence of a transition in cell fate around mid-gestation[25,73]. In the SP and CP, this transition coincided with an increase in genes expressed by radial glia (*BMP7*; *SOX3*) and oligodendrocyte precursor cells (OPCs; *CA10*) with a

**Fig. 1 | Generation of a 3D anatomical atlas of the mid-gestation foetal brain. a** Paired histological sections and simplified anatomical annotations were divided into 256 × 256 random patches (*n* = 1000) for model training. Patches were quality-checked prior to selection to ensure good overlap between labels and anatomy and no tissue damage. **b** Pix2pix model architecture showing a U-Net generator coupled with a PatchGAN discriminator. Box sizes represent image width, height and number of filters/channels (depth) at each layer. The filters and dimensions of each layer are shown below. **c** Model performance was evaluated on a set of sections that were not included in the training dataset. Checkerboard occlusions are shown with the original section, occluded patch predictions are shown using the trained model after a given number of iterations. **d** The trained model was used to replace RGB values of outlying pixels with synthetic estimates. Top row: original sections spaced throughout the cerebral hemisphere with automatically identified outlier pixels outlined in grey. Bottom row, repaired sections. **e** Repaired sections were aligned via linear, affine and iterative nonlinear registrations (see "Methods") to create a 3D volume with a final isotropic resolution of 150 μm. Right: Cut-planes illustrate

internal structures after each stage of reconstruction. The reconstructed tissue label volume is shown in Supplementary Fig. S4. VZ: ventricular zone. **f** The outer (pial) and inner (subplate) cortical plate boundaries were extracted as surface tessellations. The μBrain cortical labels were projected onto the surface vertices to form the final cortical atlas (see Supplementary Fig. S4). Cortical areas correspond to matched LMD microarray data (Supplementary Data S4 and Supplementary Fig. S4d). **g** Partial reconstructions of *EOMES*, *FOXP1* and *GRIK2* ISH data. ISH-stained sections were registered to the nearest Nissl-stained sections and aligned to the μBrain volume. Top row: selected axial and coronal sections of the μBrain volume and corresponding tissue labels with ISH expression of three developmental genes: *EOMES, FOXP1* and *GRIK2* overlaid. Expression intensity was derived from false-colour, semi-quantitative maps of gene expression. Bottom row: average expression intensity within each tissue or brain structure based on μBrain tissue labels. Averages were calculated only within sections where ISH was available for each gene. Source data are provided as a Source Data file.

transitory decrease in microglia-enriched genes in the CP (*GPR34*; *TREM2*)[74] (Fig. 2b and Supplementary Data S5).

Across all tissue samples, we tested for differences in gene expression across zones (CP, SP, etc.), regions (motor, sensory, etc) and time points (early vs mid-gestation). This resulted in a subset of *n* = 2145 (24.5%) genes with differential expression across all three factors, termed Zone-Region-Timepoint (ZRT) genes (*p* < 0.01 after FDR correction; Fig. 2c and Supplementary Data S6). We reasoned that this subset, characterised by genes with dynamic regional and temporal expression in mid-gestation, would be associated with differential rates of cortical expansion during development. To support this line of reasoning, we found that the ZRT cluster was enriched for genes upregulated in the third trimester[75] (enrichment ratio = 1.89, hypergeometric test $p_{hypergeom} < 0.0001$) and highly expressed in adolescent and adult brain tissue, compared to non-ZRT genes (Supplementary Fig. S9). ZRT genes included several human transcription factors (e.g.: *EGR1*, *JUNB*, *ZNF536*)[76] and were significantly enriched in radial glia (*SOX2*, *HES5*, $p_{hypergeom} = 0.03176$), OPCs (*OLIG1*, *PDGFRA*; $p_{hypergeom} = 0.0424$) and migrating interneurons (*CALB2*, *CNR1*; $p_{hypergeom} = 0.0009$; Supplementary Data S7)[10]. In line with a previous analysis of these data[29], we observed differential cell-type enrichments of ZRT genes across tissue zones. This reflected an expected maturational progression with genes expressed by proliferative cell types upregulated in the germinal zones and neuronal markers in post-mitotic zones (Fig. 2d and Supplementary Fig. S10). We observed highest overall ZRT expression in the subplate, with increasing expression of ZRT genes in postmitotic zones (CP, SP and IZ) compared to the SVZ and VZ, between 15 and 21 PCW (Fig. 2e). Examining ZRT gene annotations revealed enrichment of critical neurodevelopmental functions including cell-cell adhesion (GO: 0098742; *CHD1*, *EFNA5*, *NLGN1*, *NRXN1*; $p_{FDR} < 0.0001$, background set = 8771 genes), forebrain development (GO: 0030900; *CASP3*, *CNTN2*, *DLX2*, *FOXP2*, *NEUROD6*; $p_{FDR} = 0.026$) and neuron projection guidance (GO: 0097485; *EFNA2*, *EFNA5*; $p_{FDR} = 0.0034$) (Supplementary Data S8). The ZRT geneset was additionally enriched for high-confidence ASD-linked genes (*n* = 43, $p_{hypergeom} = 0.034$)[77] including *SCL6A1*, *CACNA1C* and *CHD7* and pathogenic variants in 161 ZRT (7.5%) genes have previously been linked to neurodevelopmental and cognitive phenotypes and brain malformations[78] including *MAGEL2* (Schaaf-Yang syndrome[79]), *AFF2* (Fragile-X-E[80]) and *ADGRG1* (polymicrogyria[81]) (Supplementary Data S9). Thus, ZRT genes capture a geneset enriched for neurodevelopmental functions with dynamic spatiotemporal expression in mid-gestation that continues into postnatal life, supporting a prolonged role in brain growth and development. We next sought to identify a subset of ZRT genes associated specifically with patterns of early cortical growth.

## Regional differences in the rate of cortical expansion *in utero* during the third trimester

Pioneering studies of the prenatal brain have previously demonstrated a spatial correspondence between patterns of gene expression and tissue microstructure in the foetal cortex[82]. We hypothesised that the dynamic temporal and regional patterning of ZRT genes across tissue zones would support differential rates of areal expansion across the cortex. To test this, we acquired *n* = 240 motion-corrected foetal brain MRI scans from 229 fetuses aged between 21$^{+1}$ and 38$^{+2}$ gestational weeks$^{+days}$ as part of the Developing Human Connectome Project (dHCP)[83]. Volumetric T2-weighted scans were automatically reconstructed to 0.5 mm isotropic voxel resolution[84,85], then tissue segmentations were initially extracted using neonatal protocols[86], followed by extensive manual editing to ensure accuracy ("Methods"). Manually corrected segmentations were then used to generate cortical surface reconstructions (Fig. 3a)[87]. For analysis, individual cortical surfaces were aligned to a foetal spatiotemporal atlas using a non-linear, biomechanically-constrained surface registration (Multimodal Surface Matching [MSM]; Fig. 3a–c)[88–91]. At each stage, outputs were visually quality-checked, and any failures were removed. In total, data from *n* = 195 scans in 190 fetuses (gestational age: 21$^{+1}$ - 38$^{+2}$ weeks; 88 female) were included in the analysis (Supplementary Fig. S11).

As expected, the total cortical surface area increased exponentially between 21- and 38-weeks gestation ($\beta_{age}$ = 0.054, *p* < 0.001; Fig. 3c, d)[92–94]. While cortical surface area was moderately greater in males compared to females ($\beta_{male}$ = 0.011, *p* = 0.002), this relationship did not change with age (*p* = 0.946). At each vertex in the cortical surface mesh (*n* = 30,248, excluding midline regions), we modelled areal expansion with respect to the total surface area using log-log regression (see "Methods"; Fig. 3e)[39]. Hyperallometric expansion, occurring at a rate faster than the cortical surface as a whole, was observed across the lateral neocortical surface encompassing the frontoparietal operculum and (granular) insula, primary motor and sensory cortex as well as dorsal parietal and frontal cortices, confirming previous observations based on foetal MRI and scans of preterm-born infants (Fig. 3c, e)[94–98]. In line with proposed models of cortical evolution and expansion[4,99], slower rates of growth were observed in the medial allocortex (including entorhinal, paleocortex and parahippocampal cortex) and the cingulate cortex (Fig. 3e). The inclusion of sex and sex:age interaction effects in the scaling model did not affect estimated vertex scaling coefficients (*r* = 0.996). We confirmed that estimates of cortical expansion from cross-sectional analysis aligned closely to longitudinal observations from a single foetus scanned three times during gestation (Supplementary Fig. S12).

We calculated the non-parametric correlation (Kendall's τ) between regional estimates of ZRT gene expression in the cortical plate and subjacent tissue zones and average allometric scaling in each of cortical areas defined by the μBrain atlas (Fig. 3e). In total, across

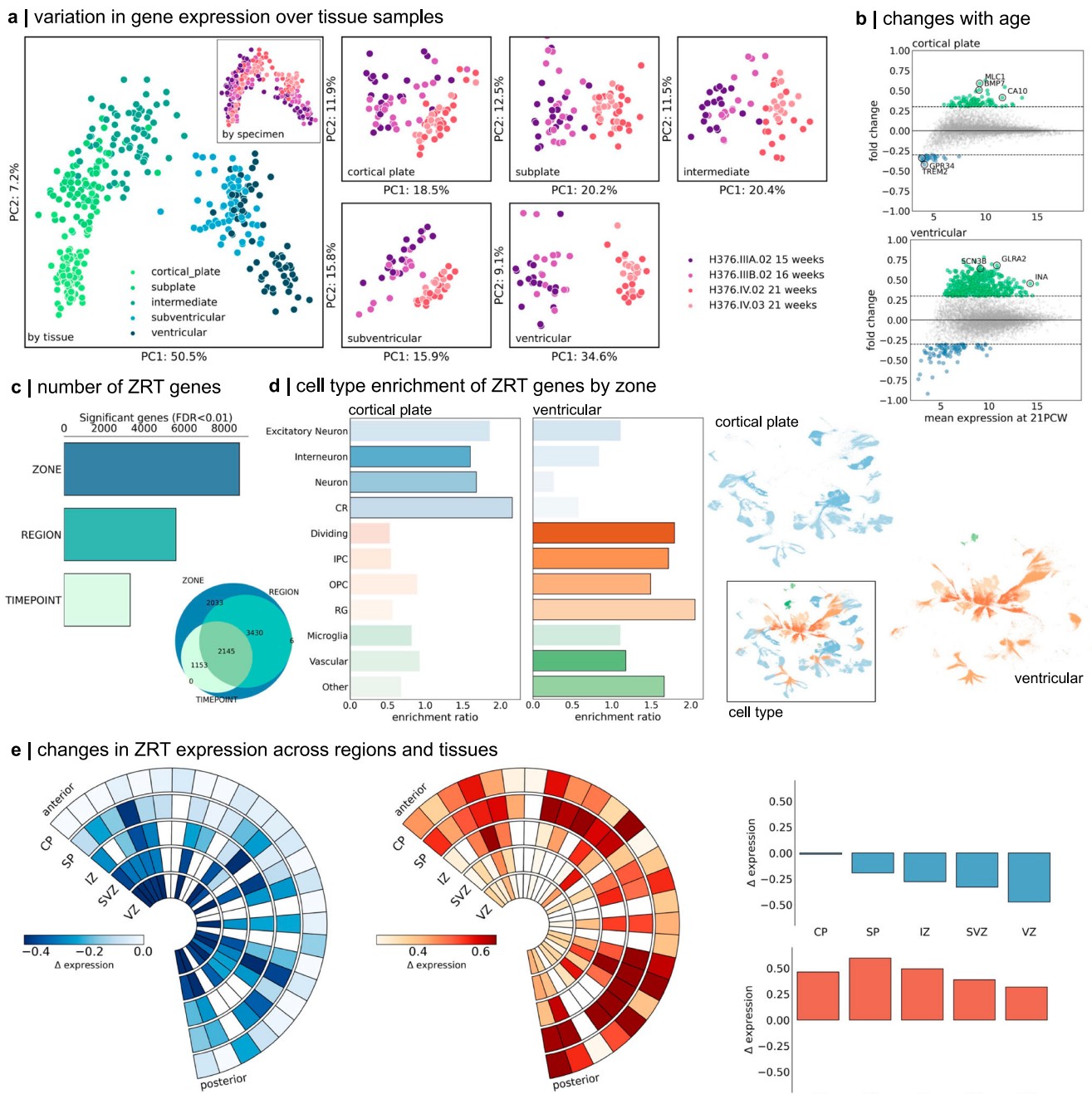

**Fig. 2 | Regional gene expression in the mid-gestation foetal brain. a** PCA of LMD microarray data (*n* = 8771 genes) in four prenatal brain specimens aged 15 PCW to 21 PCW. All tissue samples are shown (left) and coloured by tissue zones (main) and specimen (inset). PCA was applied to all samples in each tissue zone separately (right). Samples are coloured by specimen and clustered by age. PC: Principal component. **b** PC1 was associated with age-related change in all tissues and correlated between neighbouring zones. Plots show mean gene expression at 21 PCW (averaged over specimen and region) against fold change in gene expression between 15/16 PCW and 21 PCW for two tissue zones (cortical plate, top and ventricular zone, bottom). Genes with a log2(fold change) > 0.3 are shown in green (< − 0.3 in blue). Representative genes are highlighted. **c** Number of genes with differential expression over tissue zones (ZONE), cortical region (REGION) or timepoint (TIME). Venn diagram shows the overlap of gene sets. In total, *n* = 2145 were differentially expressed across zone, region and time (ZRT genes). **d** Foetal cell type enrichment[10] of ZRT genes differentially expressed in the cortical plate and ventricular zones (left). Enrichments of other tissue zones are shown in Supplementary Fig. S10. Significant cell type enrichments in each zone are highlighted with black outline (one-sided hypergeometric test; *p* < 0.05 uncorr.). UMAP projection of cell types showing enriched clusters of OPCs and radial glia in the proliferative ventricular zone, and neuronal cell types in the cortical plate. Inset: UMAP projection coloured by cell type. CR: Cajal-Retzius cells; IPC: intermediate progenitor cells; OPC: oligodendrocyte progenitor cells; RG: radial glia. **e** ZRT gene expression over time and region. Wedge plots (left) show the pattern of expression of ZRT genes that decrease (left) or increase (right) between 15 and 21 PCW. Rows indicate tissue zones, and columns indicate cortical regions ordered from anterior to posterior poles. Boxes are coloured by changes in gene expression over time (*Δ* expression). Right: bar charts show mean change in gene expression for decreasing (top) and increasing (bottom) ZRT genes averaged within tissue zones. CP: cortical plate; SP: subplate; IZ: intermediate zone; SVZ: subventricular zone; VZ: ventricular zone. Source data are provided as a Source Data file.

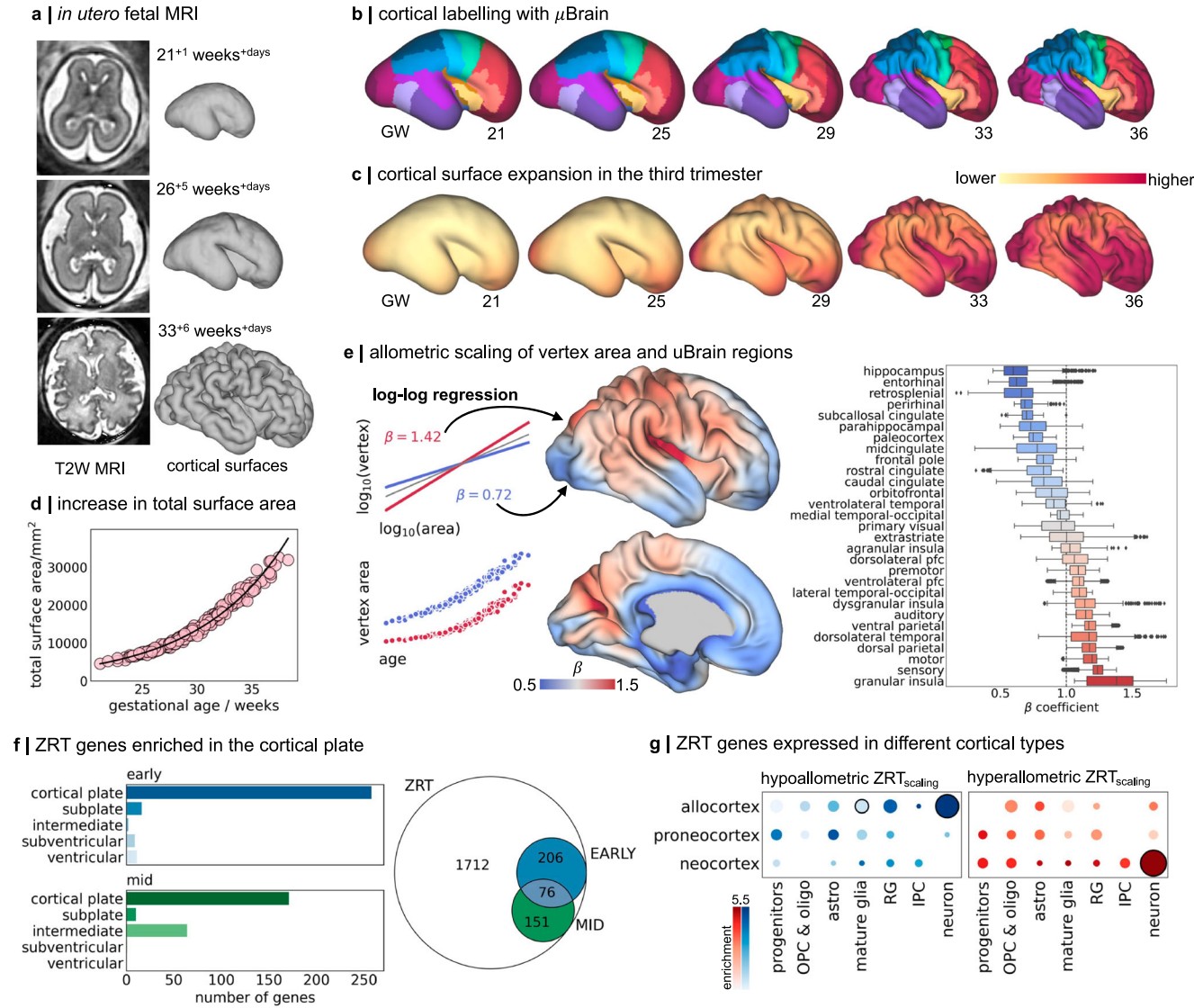

**Fig. 3 | Preferential cortical expansion during the third trimester. a** $n = 195$ foetal MRI scans were acquired during the third trimester of pregnancy. T2-weighted (T2W) scans were reconstructed using a motion-robust processing pipeline and used to generate tessellated cortical surface representations that were aligned to the dHCP foetal surface template **b** μBrain cortical labels projected onto dHCP foetal template surfaces from 21 to 36 weeks gestation using nonlinear surface registration. Surfaces are scaled to the same size for visualisation. **c** For each timepoint, weighted average vertex area maps are displayed on the respective surface templates. Foetal cortical area maps were calculated from individual, co-registered and resampled foetal surfaces using a Gaussian kernel (sigma = 1 week). **d** Total cortical surface area calculated across all surface vertices (excluding the midline) as a function of gestational age at scan. **e** Left: Models of allometric scaling were calculated for each vertex, modelling $\log_{10}$(vertex area) as a function of $\log_{10}$(total area) (top). In this framework, $\beta > 1$ indicates hyperallometric growth (a relative expansion faster than the global rate). Note that a faster growth rate does

not necessarily equate to greater total area at any given time (bottom). Middle: Hyperallometric scaling with respect to total cortical surface area ($\beta > 1$) plotted on the 36w template surface representing preferential cortical expansion during development. Right: Distribution of scaling coefficients for all vertices in each μBrain label in (**a**), ordered by mean scaling. Boxplots show quartiles (box) and range (whiskers) of areal scaling for all vertices in each region, markers indicate outliers (> 1.5 interquartile range). **f** Right: In total, the expression of 433 ZRT genes was correlated with areal scaling in gestation. Left: Significant associations (Kendall's $\tau$, $p_{FDR} < 0.01$) were observed across both early (15/16 PCW) and mid-gestation (21 PCW) time points and in all tissue zones. **g** Enrichment of hypoallometric (left) and hyperallometric (right) ZRT$_{scaling}$ genes in cortical-type specific cell markers[10]. Circle size denotes enrichment ratio, and significant associations ($p < 0.05$, one-sided hypergeometric test, uncorr.) are highlighted with black outline. OPC: oligodendrocyte precursors; Astro: astrocytes; RG: radial glia; IPC: intermediate progenitor cells. Source data are provided as a Source Data file.

both early and mid-gestation timepoints, expression of 433/2145 (20.1%) ZRT genes was spatially correlated with areal expansion during gestation in at least one tissue zone (ZRT$_{scaling}$; $n = 542$ significant associations, $p_{FDR} < 0.01$) (Supplementary Data S9, S10). Associations with areal scaling were significantly more common in ZRT genes than in non-ZRT genes (ZRT: 20.1%, non-ZRT: 8.3%; odds ratio = 2.78, $p < 0.0001$) with the most significant ZRT$_{scaling}$ associations (414/542) localised to the CP (Fig. 3f; Supplementary Data S10). ZRT$_{scaling}$ genes

in the CP included known molecular correlates of areal identity (*EFNA5*[100], *GLI3*[101], *FGFR2*[102]) and axonal guidance (*SLIT1, ROBO3, SRGAP1*)[103]. On average, expression of ZRT$_{scaling}$ genes was highest in post-mitotic tissue zones (CP, SP, and IZ; $n = 433$ genes, $\beta_{zone} = 0.68$, $p < 0.001$; Supplementary Fig. S13).

Differential expression of ZRT$_{scaling}$ genes largely captured differences between post-mitotic allocortex and neocortex, reflecting opposing allometric scaling across phylogenetic cortical types

(Fig. 3e). We found evidence at 15 PCW, but not at 21 PCW, that genes with higher expression in slower-expanding allocortex and peri-allo-cortex, were significantly enriched in early-born Cajal-Retzius neurons (e.g., *CALB2*; overlap=17, enrichment = 1.72, $p_{hypergeom} = 0.021$)[10], cells that originate from the pallial-subpallial boundary and cortical hem and migrate tangentially across the developing neocortex in early gestation[104,105]. ZRT$_{scaling}$ genes involved in Notch signalling (*NOTCH2NLR, JAG1*)[106] and others critical for hippocampal dendritic development (*LRIG1*)[107] were also expressed highly in allocortical regions (Supplementary Data S10). In contrast, ZRT$_{scaling}$ genes expressed in the preferentially expanded neocortex were enriched in progenitor cells at 15 PCW (*FBXO32, HES6*; IPC enrichment = 1.51, $p_{hypergeom} = 0.027$), and general markers of deep layer neurons at both timepoints (*NEUROD6, SYT6*; 15 PCW: Neuron enrichment = 1.53, $p_{hypergeom} = 0.004$; 21 PCW: enrichment = 1.51, $p_{hypergeom} = 0.007$). While basic cell types are generally conserved across cortical areas[108], previous evidence has shown that regional identity is imprinted during cell differentiation, with areal signatures most apparent in post-mitotic cell types but observable even at early stages of development across major brain structures[10,11,27]. In line with this, we found opposing enrichment of postmitotic allocortical and neocortical neuronal markers, but not progenitors, in hypoallometric and hyperallometric ZRT$_{scaling}$ genes, respectively (Fig. 3g).

An expanded neocortex is a hallmark of the primate brain. A recent transcriptomic survey of the neocortex across primate species identified a set of genes differentially expressed in humans (hDEGS) and located near genomic regions that are highly conserved across mammals but significantly altered along the human lineage, either through accelerated DNA substitution rates (human accelerated regions; HAR) or deletions (human conserved deletions; hCONDELS)[109–111]. We tested whether these genes were associated with human neocortical expansion in vivo. We found that ZRT$_{scaling}$ genes were significantly enriched for hDEGs located near HARs (overlap = 37; enrichment = 2.09, $p_{hypergeom} < 0.0001$) and hCONDELS (overlap = 17; enrichment = 2.0, $p_{hypergeom} = 0.008$). Of these, 22 (56%) were expressed more highly in neocortical than allocortical regions, including several cell adhesion molecules (*DSCAM, PCDH7, PCDH9, LRFN2*), teneurins (*TENM3*) and ephrins (*EFNA5*), as well as genes with functional links to language acquisition (*FOXP2*) and neurodevelopmental disorders (*MEF2C, AFF2, ZEB2*) (Supplementary Data S9).

### Prolonged neural migration precedes faster expansion across the neocortex

Focusing further on neocortical expansion, we removed allo- and transitory periallo-cortical structures (hippocampus, retrosplenial cortex, entorhinal cortex and paleocortex) and repeated our regional correlation analysis over all ZRT genes. Within the neocortex, a subset of 116 ZRT genes (including 113 ZRT$_{scaling}$ genes) were significantly associated with differential rates of expansion across neocortical regions (ZRT$_{neo}$; $p_{FDR} < 0.01$), with most associations localised to the intermediate zone (IZ; Fig. 4a and Supplemental Data S10). As with the ZRT$_{scaling}$ genes, ZRT$_{neo}$ genes were expressed more highly in post-mitotic zones than in germinal zones ($n = 116$, $\beta_{zone} = 1.11$, $p < 0.001$; Supplementary Fig. S13). Comparing ZRT$_{neo}$ genes to previously identified markers of areal identity derived from scRNA-seq[10], we observed a significant enrichment of neuron-specific areal markers (34/116 genes, enrichment = 1.34, $p_{hypergeom} = 0.037$) and, to a lesser extent, intermediate progenitor areal markers (18/116, $p_{hypergeom} = 0.045$) but not radial glia areal markers ($p_{hypergeom} = 0.257$). ZRT$_{neo}$ genes were also enriched for hDEGS located near HARs (overlap= 10; enrichment = 2.03, $p_{hypergeom} = 0.028$), including *PCDH7, PCDH9, TENM3* and *AFF2* but not hCONDELS (Supplementary Data S9).

We observed contrasting cell type enrichments of ZRT$_{neo}$ genes at 15 and 21 PCW. Consistent with the role of prolonged radial glial

proliferation in proposed models of cortical expansion[6,48,51], highly expressed ZRT$_{neo}$ genes in areas with a higher rate of expansion over gestation were enriched in radial glia and intermediate progenitors at 15 PCW ($p_{hypergeom} = 0.045$, 0.040 respectively; Supplementary Fig. S14 and Supplementary Data S11) with significant associations localised to the cortical plate, subplate and subventricular zone (Fig. 4a). Early hyperallometric ZRT$_{neo}$ genes are upregulated in both outer (*CDC42EP4, HS6ST1*) and ventricular (*FBXO32*) radial glial sub-populations (Supplementary Fig. S14)[50]. In contrast, ZRT$_{neo}$ genes expressed in neocortical areas with slower relative growth were loca-lised to the cortical plate and subplate but not specifically enriched for any major cell types (all $p_{hypergeom} > 0.05$; Supplemental Data S11). However, hypoallometric ZRT$_{neo}$ genes were expressed by neurons (*NFE2L*) and involved in dendritic (*ABGRB3*)[112] and synaptic (*NPTX2*)[113] plasticity, indicative of a population of maturing, not proliferative, cells with neuronal lineage in these regions.

At 21 PCW, after the peak period of neurogenesis, we observed the opposite pattern of cell type enrichments. ZRT$_{neo}$ genes expressed in the IZ subjacent to preferentially expanded cortical areas were enri-ched in neuronal populations (enrichment = 2.19, $p_{hypergeom} = 0.00011$) (Fig. 4a–c) whereas, hypoallometric ZRT$_{neo}$ genes were enriched in proliferative glial cell types (IPC: enrichment=2.96, $p_{hypergeom} < 0.0001$; RG: enrichment=2.36, $p_{hypergeom} < 0.0001$; Fig. 4b, c and Supplementary Data S11). The presence of post-mitotic neuronal markers in the IZ at 21 PCW suggested that neuronal migra-tion was ongoing in cortical areas with the fastest rate of expansion in the third trimester of gestation. While we cannot rule out other factors[46,47], this is consistent with a conserved mechanism of mam-malian cortical expansion whereby longer neurogenic periods lead to an expanded neocortex[6,114–117]. In this context, on both phylogenetic and ontogenetic scales, later developing cortical regions would exhibit faster rates of expansion[48,117,118]. A prominent hypothesis of neocortical expansion has suggested that, in primates, this process is realised through the continued production of upper layer neurons from outer radial glia (oRG) populations situated in the outer SVZ, a cell popula-tion greatly expanded in the primate brain[48,51]. In contrast, the expression of proliferative cell markers (Fig. 4b, c) may reflect the earlier onset of gliogenesis in hypoallometric regions.

To examine these proposed mechanisms, we focused first on *CUX1*, a marker of layer III/IV neurons that regulates dendritic morphology[119] and is expressed highly in preferentially expanded cortical regions (Fig. 4d–f). *CUX1* is located downstream of HAR426, and pathogenic mutations in *CUX1* are associated with ASD, intellec-tual disability and epilepsy[120,121]. We find that, in the IZ at 21 PCW, *CUX1* is expressed along a hypo-to-hyperallometric gradient (Fig. 4d, e; $\tau = 0.52$, $p_{FDR} = 0.002$). To validate these observations, we examine ISH staining of a second upper layer marker, *SATB2*, in five regions with differential allometric scaling, finding examples of upper layer *SATB2*$^+$ neurons within the IZ of regions with a faster rate of expansion in mid- to late-gestation (Supplementary Fig. S15). The prolonged migration of these cell populations in expanding neocortical regions is a potential consequence of differential neurogenic timing across the neocortical sheet that, at least in part, supports the accelerated expansion of hyperallometric cortical regions during gestation.

Based on this evidence, we reasoned that neuronal migration, and thus neural proliferation, in neocortical areas with slower expansion rates in the third trimester, would be complete or near complete at 21 PCW, signalling the earlier onset of gliogenesis. To test this, we com-pared ZRT$_{neo}$ genes associated with cortical scaling at 21 PCW in the IZ to region-specific cell type signatures in the mid-foetal brain[10]. Reflecting the proximity to the medial allocortex and periallocortical regions, we identified several midline identity genes (*MID1, DMRT5*) with high expression in the hypoallometric cortex as well as markers of cell proliferation (*HMMR, HAUS6, CENPN, CENPH*) (Fig. 4band Supple-mentary Data S10). In support of our hypothesis, hypoallometric

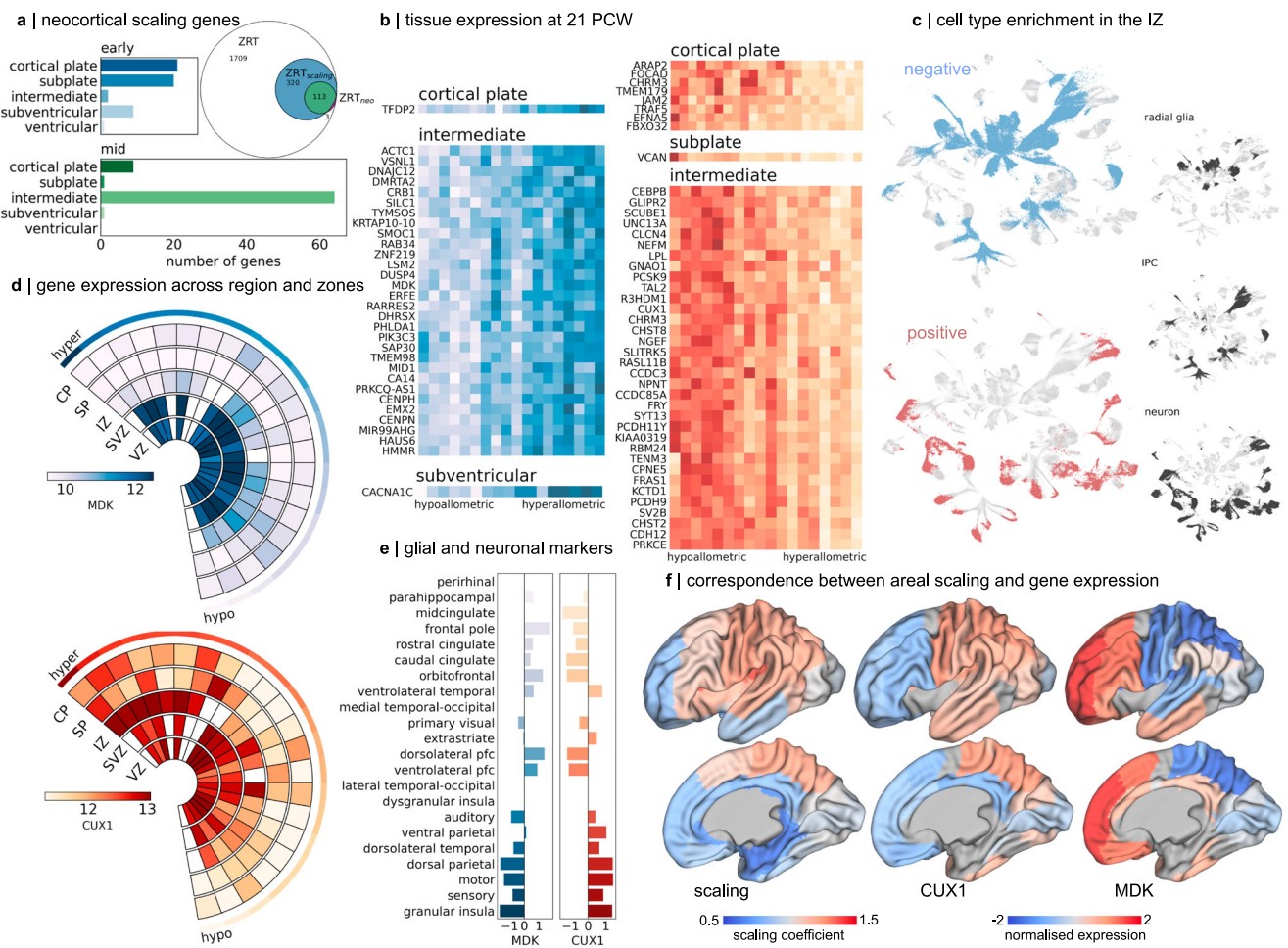

**Fig. 4 | Preferential neocortical expansion is associated with differential timing of neurogenesis and gliogenesis. a** 133 ZRT genes were associated (Kendall's tau, $p_{FDR} < 0.01$ corrected) with areal scaling of the neocortex (after excluding paleo- and archi-cortex; $ZRT_{neo}$). Most significant associations were localised to the IZ. **b** normalised (Z-score) expression profiles for genes correlated with areal scaling in each tissue zone at 21 PCW. Associations at 15 PCW are shown in Supplementary Fig. S14. Negative associations (higher relative expression in hypoallometric regions) are shown in blue, and positive associations are in red. Lighter colours indicate higher relative expression. The most significant associations are in the IZ. **c** Mid-gestation cell clusters[10] significantly enriched (one-sided hypergeometric test, $p < 0.01$ uncorr.) for genes associated with areal scaling in the IZ at 21 PCW. Territories of three cell types are shown. Negative and positive $ZRT_{neo}$ genes are enriched in progenitor cells and neurons, respectively. **d** Wedge plots are shown for two $ZRT_{neo}$ genes expressed by specific cell types: *MDK* (glial) and *CUX1* (upper layer neurons). Rows indicate tissue zones, and columns indicate cortical regions ordered according to allometric scaling from hyper to hypoallometric. The colour bar indicates normalised expression levels (a.u.). CP: cortical plate; SP: subplate; IZ: intermediate zone; SVZ: subventricular zone; VZ: ventricular zone **e** Expression (Z-score) of *MDK* and *CUX1* in all regions sampled in the IZ, ordered from hypo (top) to hyperallometric (bottom) scaling. **f** IZ expression of *CUX1* (middle) and *MDK* (right) projected onto corresponding μBrain surface atlas labels and displayed on the 36w dHCP template surface. Regions where expression for a given gene was not available are shown in grey. For comparison, the average allometric scaling in each region is displayed (left). Source data are provided as a Source Data file.

$ZRT_{neo}$ genes included recently identified markers of glial lineage (*CRB1, EMX2, SMOC1*)[122] and were specifically enriched in (peri)allo-cortical glial cell populations (*SAP30, TMEM98*; astroglia, $p_{hypergeom} = 0.01$; OPC, $p_{hypergeom} = 0.07$)[10]. *TMEM98* encodes a MYRF-interacting protein specifically expressed in newly-differentiated oligodendrocytes in the developing central nervous system[123] whereas SAP30 forms a co-repressor complex with HDAC1 and HDAC2, class I histone deacetylases that regulate gene transcription and are essential for oligodendrocyte maturation[124–127]. Similar negative correlations with cortical expansion were recorded in OPC cell population markers *S100B* (Supplementary Data S10; $\tau = -0.33$, $p_{FDR} = 0.070$), *NKX2-2* ($\tau = -0.41$, $p_{FDR} = 0.019$) and the glial progenitor marker *EGFR* (Supplementary Fig. S16), which has been validated previously as a critical gene related to brain size[128]. In an independent dataset[30], we observed similar trends in *OLIG1* expression in mid-gestation across cortical regions with differential developmental expansion (Supplementary Fig. S17). Taken together, this evidence is suggestive of an earlier onset

of glial proliferation in the IZ of hypoallometric neocortical regions in mid-gestation[129,130].

Several mechanisms exist to regulate gene transcription during early brain development[131,132]. To identify potential regulators of ZRT gene expression in the developing foetal cortex, we used recent chromatin accessibility atlas[133] to examine the position of open chromatin regions (OCR) in the mid-gestation brain relative to ZRT genes. We found that ZRT genes were more likely than non-ZRT genes to be located near predicted regulatory elements (pREs), a subset of OCRs that are likely to function as neurodevelopmental enhancers in mid-gestation[133] (OR: 1.38 $p < 0.0001$; Supplementary Fig. S18 and Supplementary Data S12). Moreover, $ZRT_{scaling}$ and $ZRT_{neo}$ genes were significantly enriched for genes located near pREs (enrichment = 1.25, 1.31 $p_{hypergeom} < 0.0001$, < 0.005 respectively; Supplementary Data S12). Focusing on laminar specificity of ZRT gene expression, we found that over 25% of $ZRT_{neo}$ genes were located immediately up- or downstream of OCRs specific to the upper layers of the cortical plate,

compared to 9% located near deep layer OCRs (Supplementary Fig. S18B). Transcription factor motifs contained within OCRs specific to upper cortical layers and proximal to $ZRT_{neo}$ ($n = 20$) included bHLH, LIM and POU homeobox and HMG-box motif families (Supplementary Fig. S18C) that bind to transcription factors which regulate superficial neuronal identify (e.g., *E2A*, *BRN1*, *LHX2*)[134–136]. Thus, the differential accessibility of specific regulatory elements can resolve the laminar identity of maturing upper-layer excitatory neurons migrating through the IZ at 21 PCW.

Overall, these data suggest that the developmental timing of the neuro-to-gliogenic switch varies across the neocortical sheet, with the length of the neurogenic period supporting differential rates of neocortical expansion during the third trimester of gestation.

### Neocortical scaling genes are critical for typical neurodevelopment

Given their likely importance in shaping early normative neurodevelopment, we hypothesised that the $ZRT_{neo}$ genes would be susceptible to severely disruptive mutations (i.e., loss-of-function variants). We found significant enrichment of hyperallometric (median loss of function observed/expected upper bound fraction (LOEUF) score = 0.26, permutation $p = 0.0003$ using random gene sets of similar size: $p_{permutation}$) but not hypoallometric (median LOEUF score = 0.40, $p_{permutation} = 1$) $ZRT_{neo}$ genes, suggesting a disproportionate level of vulnerability to loss-of-function variation in genes whose expression is greater in areas that expand fastest in the third trimester. Within these, we identified a set of constrained genes expressed highly in the subventricular zone at 15 PCW in hyperallometric regions. These genes are involved in extracellular matrix formation and interaction (*EFEMP2*, LOEUF = 0.56, *PTPRM*, LOUEF = 0.33), and epithelial-to-mesenchymal transition (*FBXO32*[137], LOEUF = 0.64), pathways crucial to outer radial glia specification and differentiation in germinal zones of the developing brain[50]. Follow-up analyses using genome-wide metrics for dosage sensitivity[138] confirmed the enrichment of hyperallometric $ZRT_{neo}$ genes as haploinsufficient (62% of genes, $p_{permutation} < 0.0001$ using random gene sets of similar size) and not triplosensitive (19%, $p_{permutation} = 0.9418$) – a highly pathogenic mechanism for loss-of-function mutations.

To assess the clinical relevance of these distinct $ZRT_{neo}$ gene sets (i.e., hypoallometric and hyperallometric), we performed enrichment analyses using MAGMA[139] across an array of previously published genome-wide association studies (GWAS). We found that $ZRT_{neo}$ gene sets were not enriched for birth outcomes (gestational duration) or cognition (educational attainment), but hypoallometric $ZRT_{neo}$ genes were enriched for externalising behaviour ($β = 0.17$, $p = 0.007$) and hyperallometric $ZRT_{neo}$ genes were enriched for schizophrenia (SCZ; $β = 0.17$, $p = 0.004$). While neither hypo- or hyperallometric genes sets were enriched for cortical thickness in adulthood, both revealed sparse enrichments for cortical surface area ($p < 0.05$, uncorr.; Supplementary Fig. S19). Further analysis using postmortem gene expression data from patients with neurodevelopmental disorders revealed significant enrichment of $ZRT_{neo}$ gene sets within multiple coexpression modules[140]. Both hypoallometric and hyperallometric $ZRT_{neo}$ genes were enriched in cross-disorder module CD1 (both $p_{permutation} < 0.05$) – downregulated in autism spectrum disorder (ASD), SCZ, and bipolar disorder, and containing neuron-enriched genes and genes with ASD- and SCZ-associated nonsynonymous de novo variants from whole-exome sequencing; and hyperallometric $ZRT_{neo}$ genes were enriched in module CD13 ($p_{permutation} < 0.05$) – also downregulated in ASD, SCZ, and bipolar disorder, and containing neuron-enriched genes.

## Discussion

Despite the altriciality of the human brain at birth, areal expansion of the cortex during the second and third trimesters of gestation is critical for later neurodevelopmental function. Cortical surface area increases exponentially during the third trimester of gestation, permitted by rapid cortical folding over the same period. Powered by a 3D atlas of the developing brain, our results provide a multiscale understanding of foetal cortical expansion in the second half of pregnancy. We find that differential expansion of cortical areas in gestation respects anatomical and evolutionary boundaries between cortical types[4,99] and is, at least in part, supported by an extended period of neural migration through mid-gestation[48,49,51].

Neurogenesis exhibits a conserved order but nonlinear scaling across species[141]. Longer neurogenic periods in larger-brained species, supported by a larger pool of progenitors in proliferative zones, result in the preferential expansion of later developing structures[6,114,115,141]. In mammals, differences in the timing and rate of neuron production vary across cortical areas with evidence to suggest that progressive termination of cortical neurogenesis occurs along a rostral-caudal axis[142–144]. In this case, earlier termination of neuronal production in the anterior cortex could create a potential affordance for increased neuronal size and arborisation, leading to increased areal expansion during development[20,141,145]. However, further evaluation of areal differences in neurogenic timing in the primate cortex presents a more complex picture, with neurogenesis terminating first in limbic and allocortical structures but continuing in the prefrontal cortex beyond mid-gestation[141,146,147]. Coupled with the nonlinear progression of human gyrification over gestation[148], this suggests that areal differences in cortical scaling are likely founded upon an alternative schema[47,48].

Alternative hypotheses have been put forward on the role of oRG proliferation and the prolonged production of neurons or glia, in cortical expansion[46–49]. Our findings demonstrate that prior to gyrification, but after the peak period of neurogenesis, supragranular neurons continue to migrate to neocortical areas with the fastest rate of expansion in the third trimester. In the primate brain, oRGs produce large numbers of upper-layer neurons, provide a scaffold for neural migration and, upon completion of neurogenesis, act as a source of glial cells in mid-to late-gestation[47,49,51,149]. Thus, regulation of neuro-to-gliogenic timing in the oRG subpopulation may represent a plausible candidate for differential rates of neocortical expansion[47,149]. Though present in other mammals, the oSVZ is expanded in primate species[48,50,51] and proliferation in the oSVZ, marked by mitotic activity, is highest in regions that expand most in later development[150]. While our data suggest rapid areal expansion is preceded by an extended neurogenic period, we lack the data to confirm a similarly extended period of gliogenesis. In humans, neurogenesis precedes cortical folding, with the subsequent gliogenic period more closely aligned to the timing of cortical expansion[46,47]. An extended neurogenic period coupled with a longer migration time due to the expanding volume of the brain may necessitate an extended gliogenic period to populate the expanding neuropil[46]. Evaluating the temporal and spatial regulation of glial fate transition and proliferation in the oSVZ during the second half of gestation represents a critical next step in understanding this process[122,151].

In gyrencephalic species, the buckling and folding of the cerebral cortex allow for increased surface area of the cortical grey matter. Greater tangential expansion of superficial cortical layers relative to subcortical tissue represents a core feature of biomechanical models of cortical growth and folding[152–155]. However, uniform rates of tangential expansion cannot fully account for the consistency in the location of cortical folds across individuals, with additional genetic contributions to gyral patterning clearly demonstrated in twin studies[156,157]. In contrast, genetically determined areal differences in expansion rate may give rise to the consistent patterns of folding observed across the neocortical sheet[150,158]. Recently, large-scale neuroimaging studies have identified patterns of altered cortical morphometry that are shared across common neuropsychiatric

conditions, and human genetics studies have begun to converge on putative mechanisms underlying cortical abnormalities in developmental genetic disorders[159–163]. Here, we identify significant enrichment of pathogenic loss-of-function variants in genes that are expressed in mid-gestation, linked to the specification of outer radial glia and associated with differential rates of cortical expansion. Taken together, these findings suggest that there are temporal windows of susceptibility in the early stages of brain development where areal differences in the timing of fundamental neurogenic processes could underlie observable cortical abnormalities and postnatal functional pathologies in neurogenetic disorders[44].

Spatially-embedded gene expression atlases of the adult human[32,66] and mouse[68,164] have proven exceptionally powerful in recent years, bridging resolution gaps to common neuroimaging modalities[165,166] and providing insight into the molecular correlates of structural[21,39,167] and functional neuroanatomy[33,168,169], brain development[170–172], disease and disorder[44,173,174]. In such studies, comparisons with in vivo neuroanatomy can only be fully realised through the three-dimensional localisation of tissue samples within a common coordinate space[60,66,68]. To date, a limitation of this approach has been either the sampling of a narrow age range outside of key developmental periods[32,67] or, in developmental datasets, a lack of 3D spatial information[52] and relatively coarse anatomical sampling[31]. To fill this gap, we provide a digital resource, µBrain, built upon existing open-source data to allow researchers to map developmental neuroanatomy of the human foetal brain onto early histogenic processes using contemporaneous post-mortem data. The reconstructed 3D µBrain atlas brings detailed tissue microarray and in situ hybridisation data into alignment with a developmental anatomical atlas of the foetal brain[88].

The time leading up to birth represents a period of highly dynamic gene expression in the human cortex[30,31,175]. In this study, our observations are limited by available microarray data to a short gestational window, precluding a fine-grained examination of concurrent changes to gene expression and cortical folding over the full third trimester. While no dataset currently exists that captures gene expression data over mid-to-late gestation at a comparable spatial resolution, the µBrain atlas provides an anatomical reference framework within which to integrate future studies. To this end, we hope that µBrain will enable future examination of tissue or region-specific expression signatures in relation to aspects of structural or functional brain development in utero or identify spatial or temporal windows of vulnerability for genetic or neurodevelopmental disorders.

Developmental MRI studies provide unique insight into early human brain development. Due to large differences in size, shape and tissue contrast, specialised tools are required for the analysis of infant and neonatal MRI. Similarly, we cannot rely on common cortical atlases that are based on adult neuroanatomy[86,176]. Prior studies have relied upon existing transcriptomic databases with relatively coarse anatomical sampling to examine spatial associations between prenatal gene expression and cortical development[30,31,98,177,178]. Here, we used annotations derived from the cytoarchitecture of the mid-foetal brain to generate a cortical atlas to facilitate further research in early brain development. Using a two-stage nonlinear registration strategy, we have aligned these labels to a spatiotemporal cortical surface atlas of the developing brain. A key area for future research in this field is the development and validation of improved methods to align early MRI to common template spaces. The geometry of the foetal cortex is smooth, making alignment of cortical morphometry an ill-posed problem. Newer, anatomically-constrained registration techniques and larger longitudinal cohorts with multiple scans during the mid to late trimester will enable more precise estimates of cortical expansion in the future[89,94].

With increasingly granular surveys of the developing brain at a single-cell level[10,108] the advent of spatial transcriptomic technologies[179]

and a series of large-scale and open-access perinatal neuroimaging studies[83,180,181], we anticipate µBrain will provide a foundation for developmental and comparative neuroscience to integrate and transfer knowledge of early brain development across domains, model systems and resolution scales.

## Methods

### Ethics
Source histological and microarray data were made available by the Allen Brain Institute https://www.brainspan.org/ using post-mortem tissue provided by the Birth Defects Research Laboratory at the University of Washington and Advanced Bioscience Resources Incorporated (Alameda, CA) with approval by the Human Investigation Committees and Institutional Ethics Committees of University of Washington. Written informed consent was obtained, and all available non-identifying information was recorded for each sample.

MRI data acquisition was performed as part of the Developing Human Connectome Project and approved by the UK Health Research Authority (Research Ethics Committee reference 452 number: 14/LO/1169). Written parental consent was obtained in every case for imaging and open data release of the anonymized data. All data was acquired at St Thomas Hospital, London, United Kingdom.

### Public data sources
Source data underlying the µBrain atlas were made available as part of the BrainSpan Developing Brain Atlas [https://atlas.brain-map.org/atlas?atlas=3] with detailed tissue processing protocols available from Ding et al. [52]. In brief, a single prenatal brain specimen (21 PCW; female) was bisected and the right hemisphere used for serial sectioning. The brain specimen was cut into four coronal slabs and frozen in isopentane. Serial coronal sectioning at 20 µm thickness was performed slab-by-slab with sequential sections submitted to Nissl, AChE or ISH staining with 43 gene probes and stained sections digitally scanned at 1 µm / pixel resolution. In total, 81 out of 174 Nissl-stained sections with varying sampling densities (~0.5 mm to 1.2 mm between sections) were selected for annotation[52]. Expert anatomical annotations were conducted manually on each section. Nissl- and ISH-stained sections with corresponding anatomical labels were made available for download. Anatomical annotations were also used to guide laser microdissections for DNA microarray analysis across the developing cerebral tissue in the left hemisphere of 4 separate mid-gestation specimens (see Microarray Data below). The section numbers and approximate coronal positions of sections used in the construction of the 3D µBrain atlas are listed in Supplementary Data S1.

### Image processing
We downloaded each high-resolution Nissl-stained section ($n = 81$; downsampled to 2µm/pixel) as RGB images in JPG format with corresponding anatomical labels as SVG files.

After converting SVG to RGB PNG format, we manually combined anatomical labels according to the hierarchical ontology of the reference atlas[52] to create two compact annotations, one for image repair comprising 20 tissue structure labels (brain-labels) and one for statistical analysis containing only cortical labels (cortex-labels, $n = 30$, including one generic 'brain tissue' label for non-cortical structures; see Supplementary Data S2). Due to the small size and degree of missing data precluding reconstruction, marginal zone and subpial granular zones were not considered in this analysis. Nissl-stained sections and corresponding label images were then downsampled to 20µm /pixel resolution.

### Histological reconstruction
Pix2pix is a conditional generative adversarial network (GAN) trained to perform image-to-image translation between pairs of image

examples[57]. We used the pix2pix architecture (Fig. 1b) to synthesise Nissl-stained images from label annotations in order to replace artefacts within tissue sections (Fig. 1c, d).

**Model architecture.** Following the conventional GAN structure, the model combines a generator network, *G*, with a classifier (or discriminator, *D*) with an objective to determine if images are real or fake (Fig. 1b). Following Isola et al. [57], our generator network takes the form of a U-Net, with a symmetric encoder-decoder structure, and skip connections between corresponding encoding and decoding paths. The encoder was parameterised with 7 resolution levels, each comprising a 2D convolution [kernel size = (4,4), stride = (2,2); filters = (64, 128, 256, 512, 512, 512, 512)], batch normalisation and a leaky ReLu activation function ($\alpha = 0.2$). This is followed by a bottleneck layer composed of one 2D convolution with 512 filters and ReLu activation. The decoder layers consisted of 2D upsampling (implemented with nearest neighbour interpolation), followed by a 2D convolution [kernel size = (4,4), stride = (1,1), filters = (512, 512, 512, 512, 256, 128, 64)] with dropout ($p = 0.5$) (Fig. 1b). A final upsampling and convolution with tanh activation was applied to generate the 3-channel RGB output image.

The pix2pix discriminator, *D*, is trained as a convolutional Patch-GAN. By acting on small sections of an image rather than the whole image, this approach focuses on high-frequency image structure and texture while requiring fewer parameters[57]. In our application, the discriminator comprised three 2D convolutional layers [kernel = (4,4), stride = (2,2), (1,1) and (1,1), filters = 64, 128 and 1, respectively] with leaky ReLU (first two layers) and sigmoid (final layer) activations (Fig. 1b). Batch normalisation was applied after the second convolution. The final output is a 128 × 128 image with real/fake predictions for each 16 × 16 patch in the original input image[57]. We implemented the model in Python (3.7) using Tensorflow (2.4.1).

**Training data.** We trained the pix2pix model on 1000 pairs of 256 × 256 image patches from 20μm resolution Nissl-stained sections and corresponding label annotations (Fig. 1a, b). Each pair of training patches was visually inspected to ensure no tissue artefacts were present in the histological data, and good alignment was observed between the tissue section and anatomical labels. Pairs that failed visual inspection were rejected and replaced until 1000 pairs were selected. Pairs were automatically excluded if > 75% of pixels were labelled as background. Patches were randomly sampled from *n* = 73 sections, with the remaining *n* = 8 sections forming a validation set to evaluate model performance. Validation sections were spread evenly through the cerebral hemisphere. To increase the size and diversity of our training set and limit overfitting, we applied common data augmentation steps including random crop, jitter and flipping of images during training.

**Model training.** GAN training is performed adversarially, with the generator network competing to generate more and more realistic synthetic images from label annotations, and the discriminator working to discriminate between real and fake examples. The model was trained in steps, alternating between training the generator and the discriminator with real and synthetic samples. As in Isola et al. [57], the discriminator loss was divided by 2 to slow down its learning rate compared to the generator and both *D* and *G* were trained using the Adam optimiser ($\beta_1 = 0.5$, $\beta_2 = 0.999$) with a learning rate of 0.0002 and batch size of 1 for a total of 100 epochs. We set the regularisation parameter, λ, to 1. Alternative parameter settings are explored in Supplementary Fig. S2. Finally, the 3-channel RGB label images used for training were transformed into 21-channel images (20 tissue labels + 1 background label), with each channel containing a 1 in pixels belonging to a given label (0 otherwise).

**Model evaluation.** We retained eight full histological sections spread evenly through the cerebral hemisphere as a validation set. No image patches used for training were drawn from this dataset. To validate model performance, we split each Nissl-stained section and its corresponding anatomical label image into non-overlapping 256 × 256 patches. Labelled patches were used to generate new synthetic Nissl-stained patches by the trained generator.

To quantify the visual and textural similarity between synthetic and ground truth Nissl images, we converted each image patch from RGB to HSV (Hue, Saturation, Value) format and calculated the similarity between hue and saturation values across both patches as:

$$\text{sim}(F_1, F_2) = \frac{1}{1 + \sqrt{\sum (F_1 - F_2)^2}} \qquad (1)$$

Where $F_1$ and $F_2$ are the vectors of pixel hue or saturation values for ground truth and synthetic image patches, respectively, normalised to unit length. As an additional measure of generator performance, we calculated a 'perceptual' similarity between ground truth and synthetic image patches based on high-level features of a large image recognition model pretrained on the ImageNet dataset (VGG19)[182–184]. After removing the fully-connected classification layers of the pretrained VGG19 model, each image patch (real and synthetic) was passed through the network, resulting in a 512-length vector output by the final layer to act as a high-level feature representation of each patch. Similarity between feature vectors was calculated as above, baseline measures of hue, saturation and perceptual similarity were calculated after randomising pixels within each ground truth patch.

**Image repair.** To perform the repair of whole sections, we split each labelled image into patches of 256 × 256 pixels with an 8 pixel overlap and passed them through the trained generator. The resulting synthetic Nissl contrast patches were stitched together into a full section matching the dimensions of the original image (Fig. 1d). Patch prediction and image reconstruction were performed using MightyMosaic (1.2.3) [https://pypi.org/project/MightyMosaic/].

To detect regions of the original Nissl-stained section that needed repair, we designed an automated outlier detection method based on the Median Absolute Deviation (MAD) of pixel hue and saturation. The original Nissl-stained sections and corresponding GAN-generated predictions were transformed to HSV format and blurred with a box filter (width = height = 5 pixels). We identified outliers with median absolute differences in hue and saturation between pixels in the ground truth image and its synthetic equivalent greater than threshold, θ, set to 2.5, whereby lowering θ would increase the number of pixels marked as outliers.

For each section, a binary mask was created containing all pixels identified as outliers in both hue and saturation. A final opening operation was applied to the outlier mask using an elliptical filter (iterations = 3, width = 3 pixels) to remove speckles in the mask. Identified outlier pixels were then replaced with the corresponding, intensity-matched pixels from the synthetic image using Poisson image editing to effect image repair (Fig. 1d)[59]. Outlier detection and repair were performed in Python (3.7) using OpenCV (4.5.2) [https://opencv.org/].

## μBrain volume construction

Following automated repair of major tissue artefacts present in the histological data, we aimed to develop a 3-dimensional reconstruction of the foetal brain to facilitate comparison with in vivo MR imaging data. Image alignment and reconstruction steps are summarised below.

**Slice-to-slice alignment.** Using the middle section as a reference, repaired Nissl-stained sections were aligned using a graph-based, slice-to-slice registration[185,186]. Repaired Nissl-stained images were

converted to greyscale and padded with a 200-pixel zero-filled border to allow for large translations while retaining the section in the field of view during image registration. Corresponding label images were converted from RGB format to one-channel label images, with labels numbered according to a look-up table (Supplementary Data S2).

For initial alignment, we implemented a graph-based, slice-to-slice registration to a chosen reference section via shortest-path transforms[185,186]. The central section was chosen as the reference, and pairwise rigid transforms were estimated between each section and up to five neighbouring sections in the direction of the reference. After each transform, a weight is calculated between the aligned sections:

$$\omega_{i,j} = (1 - r_{i,j}) \times (1 + \lambda)^{|I_i - I_j|} \qquad (2)$$

Where the weighting, ω, between two aligned images, $i$ and $j$, depends upon their pixelwise correlation, $r$, after alignment weighted by the number of intermediate sections between them (based on the absolute difference of section index, $I$). The hyperparameter, $\lambda$, was set to 0.5 and acts as a penalty on skipping slices, with a higher value penalising transforms that skip intermediate sections. Dijkstra's shortest-path algorithm was then used to calculate the set of transforms with the lowest cost to align a given section to the refs. 185,186. The selected transforms were composed and applied to both the image and its corresponding labels to bring all sections into approximate alignment (Fig. 1e and Supplementary Fig. S20a).

Aligned sections were stacked along the anterior-posterior axis into a 3D array with a resolution of 0.02 × 0.02 × 0.5 mm. Section thickness was determined by the sampling strategy of the original data (detailed in Ding et al. [52]). Consecutive sections included in the atlas were spaced approximately 0.5 mm apart (Supplementary Data S1) and were thus assigned a nominal thickness of 500 μm. Due to the sampling strategy employed during tissue sampling and histology, tissue sections were sparsely sampled along the length of the cerebral hemisphere. Where full sections were missing or excluded, or the distance between adjacent slices was larger than 0.5 mm, we repeated the preceding sections up to 5 times to fill the gap. This strategy was selected to limit abrupt transitions between neighbouring sections, preserve the overall volume of the hemisphere and allow for volumetric registration in subsequent steps. To account for missing slices at the anterior and posterior poles, manual labels were drawn following the tissue contours of the adjacent slices to create a synthetic cortical label. The trained pix2pix generative model was then used to generate a Nissl-stained coronal section to append to each pole. This process resulted in a 3D NIFTI volume of voxel size 1420 × 2678 × 125 and voxel dimensions 0.02 × 0.02 × 0.5 mm (Fig. 1e and Supplementary Fig. S20).

**Affine registration to a foetal brain shape reference.** Reconstructing 3D volumes from the consecutive alignment of 2D sections commonly produces an artefact termed: z-shift, caused by the propagation of registration errors between adjacent slices and resulting in a distorted three-dimensional structure in the final volume[187]. To overcome this effect, it is common to use a shape prior to guide registration and preserve 3D shape[60,187,188]. In lieu of a ground-truth volume for the sectioned data, we employed a population-based average anatomical image: specifically the 22-week timepoint of the Gholipour et al. spatiotemporal foetal MRI atlas (Supplementary Fig. S3)[61].

The Gholipour MRI atlas contained T2-weighted anatomical foetal MRI templates and a set of 50 anatomical brain tissue labels at 1 mm isotropic voxel resolution. We downloaded the T2-weighted template image and accompanying tissue labels for the 22-week timepoint, removing extracerebral CSF, midbrain and cerebellar structures and matching tissue labels of the MRI atlas to the μBrain tissue labels. We upsampled the MRI template and labelled images to 50 μm isotropic resolution before cropping and rotating into approximate alignment

with the 3D μBrain volume. We converted the MRI volume into a Nissl-like contrast using the trained GAN model by slicing the label image coronally and passing each slice as input to the trained model (Supplementary Fig. S3c, d). Nissl-contrast images were re-stacked into a 3D volume and used as an anatomical prior for registration.

We performed an iterative affine registration procedure between the MRI-based shape prior and the 3D stack of histological sections by first estimating an affine alignment of the MRI-based anatomical prior to the histological volume[185]. The transformed anatomical prior was then resliced, with each coronal section acting as a target for 2D registration with the corresponding histological section. The registered histological sections then form the target for the next 3D registration (Supplementary Fig. S20). This process was repeated for a total of 5 iterations, producing a final 3D volume with aligned coronal slices and a global shape approximately matched to the in-utero foetal brain (Fig. 1e and Supplementary Fig. S20a). A final 2D affine registration was calculated between the original and final aligned histological sections, and the Nissl-contrast images and corresponding tissue labels were transformed into the 3D volume.

**Final template construction.** Typically, brain templates are probabilistic estimates constructed from multiple individual datasets, representing a population-average anatomy, the principal benefit of which is to provide a common coordinate space for analysis and remove bias towards any individual's brain anatomy[189]. Borrowing from this philosophy, we framed the output of the preceding steps: a 3D volume with approximately aligned coronal sections and global shape, as a single possible representation of the ground truth cerebral volume that captures various idiosyncrasies of the reconstruction pipeline, including tissue sectioning frequency and selection, image repair, registration and/or potential misalignment. To create the final 3D volume, we employed a data augmentation technique, generating $n = 50$ unique representations of the affinely aligned data by deleting and/or repeating up to 25 randomly selected slices along all three image axes. Matched augmentations were applied to both the histological reconstruction and accompanying anatomical labels (Supplementary Fig. S20).

For each volume, we resampled the in-plane, coronal resolution to 150 × 150 μm resolution (slice thickness = 500 μm) and, for each coronal slice, performed a weighted nonlinear registration to neighbouring sections (symmetric normalisation [SyN] metric=cross-correlation; flow_sigma = 3.0, total_sigma = 1.0, grad_step = 0.25). Adjacent sections (up to 8 neighbours) were weighted based on distance to the source section. After registration, the halfway transform was applied to each section. Section-to-section registration and transformation were performed once in the anterior-posterior direction, before repeating in the posterior-anterior direction. For each volume, slice-to-slice nonlinear registrations were calculated for a total of 3 iterations.

Finally, to create a smooth 3D reconstructed volume, we co-registered all 50 augmented and aligned volumes into a single probabilistic anatomical template with voxel resolution 150 × 150 × 150 μm using an iterative, whole-brain nonlinear registration (SyN metric=cross-correlation; iterations=3; grad_step = 0.25; Fig. 1e and Supplementary Fig. S20a). Transforms were applied to each of the corresponding anatomical label volumes, and a majority vote was used to create a final set of brain tissue and cortical labels (Supplementary Fig. S4). All image registration was performed in Python (3.7) using antspyx (0.2.7)[190].

**Cortical reconstruction.** To reconstruct the foetal cortical surface, we adapted existing protocols for ex vivo [https://freesurfer.net/fswiki/ExVivo] and non-human primate [https://prime-re.github.io/] surface reconstruction with Freesurfer[191]. We used the μBrain tissue labels to generate a 'white matter' mask (all subcortical structures and tissue zones, excluding the cortical plate). This volume was tessellated

(Freesurfer commands: mri_pretess, mri_tessellate), and the initial surface smoothed (mris_smooth) and inflated to a sphere (mris_inflate, mris_sphere). Topological errors in the initial surface estimates were detected and fixed using manual edits to the brain and white matter masks, before repeating the process. Finally, a pseudo-T2 volume was created using tissue labels, assigning all voxels in the cortical plate intensities expected in grey matter by Freesurfer. This volume was used to generate inner and outer surfaces for the volume (mris_make_surfaces). (Fig. 1f). Surfaces were smoothed for 50 iterations and inspected for topological errors before conversion to gifti format and rescaling to the original size (Fig. 1f and Supplementary Fig. S4). All processing was performed with Freesurfer (7.3.2).

### In situ hybridisation

In addition to serial Nissl staining, interleaved coronal sections were used for in situ hybridisation (ISH) of a series of neurodevelopmental marker genes (Supplementary Data S3)[52]. High-throughput ISH staining was performed for each gene, with stained sections digitised at 1 μm resolution. Quantification of the intensity of expression detection was performed using an automated procedure that pseudo-colour coded levels of expression for visualisation, with low-to-high expression represented as blue-to-red[164].

Compared to Nissl-stained sections ($n = 79$ after quality control), fewer ISH-stained sections were available for each gene (mean $n = 41$ after quality control), precluding a full 3D reconstruction of each. We downloaded each set of ISH-stained sections and removed any with large artefacts (tearing, folding, missing tissue). From each false-colour expression map, we extracted the red channel to focus only on higher-expressing cells. Each section was registered to the nearest, repaired Nissl-stained section using affine registration. Registrations were visually inspected, and any failures were removed. Aligned sections were then stacked together, with blank slices in place of missing sections and reconstructed into a 3D volume using the previously calculated slice-to-volume alignments for each section.

### Microarray data

We downloaded prenatal LMD microarray data from the BrainSpan database [https://www.brainspan.org/]. For details on tissue processing and dissection, see Miller et al. [29]. In total, normalised microarray data from 58,692 probes in 1206 tissue samples were available to download, obtained from the left hemisphere of four post-mortem foetal brain specimens (age 15-21 PCW, 3 female)[29]. Each probe was assigned a present or absent annotation based on the strength of the average probe expression over the corresponding background signal. Through comparison with the BrainSpan reference atlas, we matched each tissue sample's anatomical label to (i) corresponding cortical labels included in the μBrain atlas and (ii) one of five tissue zones (cortical plate, subplate, intermediate zone, subventricular zone, ventricular zone) (Supplementary Data S4 and Supplementary Fig. S4). Samples that could not be matched to labelled regions in the cortical plate or corresponding subjacent tissue zones were removed, including samples from subcortical nuclei, midbrain structures and brainstem.

**Microarray processing.** We updated gene assignments for the Allen microarray probes using Re-Annotator[192] and removed any probes assigned to more than one gene, resulting in a probe set ($n = 46,156$) mapped to 20,262 unique genes. Low signal probes designated absent were removed (34.67% of probes), as were tissue samples from the marginal zone, subpial granular zone and subcortical and midbrain structures (54.46% of samples). Where multiple probes mapped to a single gene, the probe with the highest differential stability (DS)[193], the average pairwise correlation between tissue sample expression over all specimens, was assigned. Probes with DS < 0.2 were removed.

Where more than one sample was available for a given region or zone, e.g., samples from the outer and inner cortical plate in the same region, gene expression was averaged across samples. Finally, any probes with missing data in more than 10% of tissue samples were removed ($n = 1253$). This resulted in expression data from 8771 genes across 27 regions and 5 tissue zones for analysis (Supplementary Fig. S4).

### Foetal MRI

To measure cortical expansion in utero during the third trimester, we analysed high-resolution MRI from a large cohort of foetuses.

**MRI acquisition.** Foetal MRI datasets ($n = 240$ scans from 229 foetuses aged between $21^{+1}$ and $38^{+2}$ gestational weeks$^{+days}$) were acquired as part of the Developing Human Connectome Project (dHCP) using a Philips Achieva 3 T system, with a 32-channel cardiac coil in the maternal supine position. Structural T1-weighted (T1w), T2w, functional MRI and diffusion MRI data were acquired for a total scan time of approximately 45 min[84]. T2-weighted SSTSE volumes were acquired with TE = 250 ms, acquisition resolution $1.1 \times 1.1$ mm, slice thickness 2.2 mm, −1.1 mm gap and 6 stacks. All 3D brain images were reconstructed using a fully automated slice-to-volume reconstruction (SVR) pipeline[85] to 0.5 mm resolution and reoriented to the standard radiological space.

After image processing and quality control, the final dataset comprised $n = 195$ foetal MRI datasets acquired from $n = 190$ fetuses aged $21^{+1}$ to $38^{+2}$ gestational weeks (88 female). Repeated scans were acquired from four foetuses.

**MRI processing.** While neonatal protocols for automated MRI tissue segmentation exist[86,194], due to the differences in size, tissue contrast and signal-to-noise ratio, segmentations derived from foetal MRI often require extensive manual editing to ensure accuracy[195].

Here, we used an optimised neonatal tissue segmentation pipeline (Draw-EM)[86] with tissue priors adapted to a foetal MRI template to create a 'first-pass' tissue segmentation for each foetal MRI volume. Tissue segmentations were then visually checked, and extensive manual corrections were performed where needed to correct gross segmentation errors and ensure the accuracy of tissue boundaries (CSF/cortex/white matter). Manually corrected tissue segmentations were then used to generate anatomically and topologically correct inner and outer cortical surfaces using Deformable[87]. Note that all intensity-based correction terms were turned off during surface reconstruction, and each surface was generated using just the corrected tissue segmentations. At each stage, images and derived outputs were visually inspected for accuracy.

**Alignment to foetal template.** We aligned individual cortical surfaces to the dHCP foetal atlas, a spatiotemporal surface atlas, spanning 21–36 weeks of gestation with weekly timepoints[88,90]. Using MSM with higher-order clique reduction, we calculated non-linear transforms of individual surfaces to their closest foetal timepoint based on spherical registration of sulcal depth features[89,91]. The MSM transform was used to resample individual surface topology (pial, midthickness, and white) onto the template surface vertices, ensuring that all surfaces across individuals had the same vertex correspondence. Resampled surfaces were manually checked to ensure the quality of the registration.

**Alignment to μBrain.** We aligned the μBrain cortical surface to the earliest timepoint of the dHCP foetal template surface using a two-step nonlinear surface registration guided by a set of anatomical priors (Supplementary Fig. S20b, c). We used MSM to perform an initial nonlinear spherical registration between μBrain and dHCP surfaces based on the alignment of sulcal depth. After this, we created a set of coarse cortical labels on the dHCP surface matched to corresponding

µBrain labels by combining (a) dHCP cortical atlas labels[86], (b) manual labels guided by sulcal anatomy on the 36-week foetal surface and (c) combining µBrain labels in the same lobes (e.g., ventrolateral frontal, dorsolateral frontal, orbitofrontal) into single anatomical labels. The full list of 11 matched cortical regions included: the auditory cortex, cingulate cortex, frontal cortex, insular cortex, primary motor, primary sensory, occipital cortex, parahippocampal cortex, parietal cortex, superior temporal cortex, ventrolateral temporal cortex. A secondary multivariate spherical registration between µBrain and foetal surfaces was initialised using the previously calculated sulcal alignment and driven by the alignment of cortical ROIs across surfaces[89]. This approach leverages anatomical labels (defined based on cytoarchitecture, or using older foetal anatomy in µBrain and dHCP atlases, respectively), to inform cortical alignment in the absence of geometric features. A similar approach has proven successful in accommodating large deformations across primate species[196].

µBrain labels were propagated to each timepoint of the dHCP foetal atlas (Fig. 3b) and onto the surface topology of each foetal scan. Cortical labelling was visually quality checked for alignment.

## Statistical analysis

**Allometric scaling of cortical surface area.** Each subject's outer cortical topology was resampled onto the dHCP template surface (32,492 vertices), and vertex-wise estimates of cortical surface area were corrected for folding bias by regressing out cortical curvature[197,198] and smoothed with a Gaussian kernel (FWHM = 10 mm). The total cortical surface area was calculated as the sum of all vertices in the cortical mesh, excluding the medial wall. At each vertex, $v$, we modelled scaling relationships with brain size by estimating the log-log regression coefficient for the total surface area as a predictor of vertex area, $a_v$:

$$log_{10}(a_v) = 1 + \beta log_{10}\left(\sum_{v=1}^{V} a\right) + \varepsilon \quad (3)$$

Such that the scaling coefficient, $\beta$, can be directly interpreted relative to 1 (representing linear scaling between vertex area and total area) with $\beta > 1$ and $\beta < 1$ representing hyper- and hypoallometric scaling of vertices with respect to total area, respectively[39]. Models were fit using Ordinary Least Squares (OLS) regression. We tested alternative models, including sex and age:sex interactions. Analyses were repeated after removing repeated scans to satisfy i.i.d. assumptions of OLS regression ($n = 190$; Supplementary Fig. S21).

Prior to analysis, vertexwise outliers were identified and removed (Supplementary Fig. S22). To account for age-related increases in area, outliers were identified using a sliding window over age (outliers > 2.5 S.D. from the mean within a given window, maximum window size = 25 scans, sorted by age). Data from five scans were removed prior to analysis due to the presence of outliers in more than 5% of vertices.

Vertexwise maps of areal scaling ($\beta$ coefficients) were parcellated using the µBrain cortical labels, calculating average scaling within each parcel for further analysis.

**Modelling changes in gene expression over zone (Z), region (R) and time (T).** For each gene ($n = 8771$), we modelled the main effects of cortical tissue zone, region and timepoint on expression using a general linear model. Significant effects ($p < 0.01$) were identified after False Discovery Rate correction for multiple comparisons over genes. Linear models were also used to test mean differences in average ZRT gene expression between post-mitotic and germinal zones. Statistical analysis was performed with statsmodels (0.13.5) and scikit-learn (0.24.2).

**Enrichment analyses.** For all enrichment analyses, we calculated the enrichment ratio as the ratio of the proportion of genes of interest within each geneset/marker list to the proportion of background genes within each geneset. Unless otherwise stated, the background set was defined as the full list of genes included in the study ($n = 8771$). Significance was determined using the hypergeometric statistic:

$$p = 1 - \sum_{i=0}^{x} \frac{\binom{K}{i}\binom{M-K}{N-i}}{\binom{M}{N}} \quad (4)$$

Where $p$ is the probability of finding $x$ or more genes from a specific geneset $K$ in a set of randomly selected genes, $N$ drawn from a background set, $M$. Where stated, False Discovery Rate (FDR) correction was applied to multiple comparisons.

## Reporting summary

Further information on research design is available in the Nature Portfolio Reporting Summary linked to this article.

## Data availability

The µBrain digital template with corresponding cortical surfaces, atlas labels and processed microarray data used in this study is available from https://garedaba.github.io/micro-brain and is deposited at https://doi.org/10.5281/zenodo.10622336. All dHCP data, foetal brain reconstructions, brain region segmentation and cortical surfaces are available for download from the NDA https://nda.nih.gov/edit_collection.html?id=3955. Source histological and microarray data are available from the Allen Brain Institute https://www.brainspan.org/ Source data are provided with this paper.

## Code availability

Code supporting data processing and analysis for this manuscript is available at: https://github.com/garedaba/micro-brain and is deposited at https://doi.org/10.5281/zenodo.13917290.

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

## Acknowledgements

This research was supported by the National Health and Medical Research Council (NHMRC) [1194497 to G.B.], the Murdoch Children's Research Institute, the Royal Children's Hospital, Department of Paediatrics, The University of Melbourne and the Victorian Government's Operational Infrastructure Support Programme. The project was generously supported by RCH1000, a unique arm of The Royal Children's Hospital Foundation devoted to raising funds for research at The Royal Children's Hospital [M.L.S., G.B]. L.Z.J.W. was supported by the Commonwealth Scholarship Commission, United Kingdom. V.Ka. was supported by the MRC (UK) [MR/V036874/1] and The Developing Human Connectome Project. E.C.R. was supported by a Wellcome Collaborative Award (215573/Z/19/Z). A. A.-B. and J.S. were supported by R01MH132934. Neuroimaging data were provided by the Developing Human Connectome Project, KCL-Imperial-Oxford Consortium, funded by the European Research Council under the European Union Seventh Framework Programme (FP/2007-2013) / ERC Grant Agreement no. [319456; A.D.E., J.V.H.]. We are grateful to the families who generously supported this trial. We are grateful to the Allen Institute and associated Investigators for the provision of the Atlas of the Developing Human Brain.

## Author contributions

G.B., S.O. V.Ky., L.Z.J.W., V.Ka., A.P., J.V.H., J.H., E.C.R. and J.S. performed data acquisition and data processing. G.B., S.O. and J.S. performed data analysis. G.B., J.S., V.Ky., L.Z.J.W. and E.C.R. contributed to the methodology. G.B., M.L.S., A.A.-B., J.V.H., A.D.E., E.C.R. and J.S. provided resources and supervision. E.C.R. provided software. Project conceptualisation: G.B and J.S. Writing draughts, revisions and editing: all authors.

## Competing interests

J.S. and A.A.-B. are co-founders of Centile Bioscience. The remaining authors declare no competing interests.
