## [Transparent Peer Review file · Nature Communications]

Molecular signatures of cortical expansion in the human fetal brain

Corresponding Author: Dr Gareth Ball

Version 0:

Reviewer comments:

Reviewer #1

(Remarks to the Author)

This is an interesting manuscript that imputes gene expression data from previously acquired laser capture micro-dissection of the developing human fetal cortex into a 3D atlas, fixing errors in 2D sections through deep learning based imputation. Then, regional gene expression is compared with regional expansion of the brain during development. 116 genes are identified that show a spatial correlation with regional cortical expansion. The genes in fetal development that are spatially correlated with areal expansion are also enriched in genes associated with schizophrenia from GWAS, which is interesting. I think this resource would be widely used for comparison of fetal brain gene expression with fetal and adult MRI, and therefore support the publication. However, I have several concerns about the analyses and interpretations.

1. It is surprising in Figs 3f and 4a that genes in the VZ or SVZ (which contain progenitors) have very little spatial correlation with expansion of that cortical region. Most hypotheses about cortical expansion concern progenitor fate decisions, as are discussed here. In Figure 3g and 4c the authors show an enrichment of hyperallometric ZRT_scaling or ZRT_neo within progenitors (though less than neurons). This does not seem to match the findings in Figure 3f and 4a because very few genes are spatially correlated with expansion in the zones where progenitors are actually found. How do the authors interpret this finding? Progenitor genes important for regional expansion are correlated with regional expansion not in the progenitors but instead the neurons of the cortical plate where no progenitors exist? These inconsistent findings made the biological conclusions hard to understand. It seems to me that an alternative approach is rather than correlating the level of gene expression within a specific region to expansion of that region, instead correlate the log2FC of the difference between neuronal (CP, SP, IZ) and progenitor containing (VZ, SVZ) for each region with the expansion of that region. Perhaps then the results would be more interpretable.

2. The comparison of genes that are differentially expressed across all zones, time and region with cell types (Figure 2d) conflates these three important variables, and makes the results impossible to interpret. Additionally, the lack of selection for genes upregulated or downregulated again makes the results difficult to interpret. I think it would be more useful to separately compare differentially expressed genes across zones, across regions, and across timepoints with cell types and gene ontologies where there are clear expectations for the outcomes. For example, genes higher expressed in CP > VZ should be neuronal and enriched in neuron populations. I did not find Figure 2e helpful in interpretation because this shows the differences in gene expression within the ZRT genes, but not the function of the genes.

3. In Figure 1g it would be helpful to have a more zoomed in representation of the ISH reconstructions. It is surprising that EOMES is not consistently expressed across the SVZ of the dorsal cortex and only found in the ventral cortex.

4. ZRT genes are labeled as Zone-Region-Tissue in the text, but I believe it should be Zone-Region-Timepoint to match Figure 2c.

5. It would especially be interesting if ZRT_neo overlapped with known signaling pathways involved in areal identity in the cortex or known areal markers from scRNA-seq (<https://www.nature.com/articles/s41586-021-03910-8>)

6. With the lack of cell type specific resolution of the LCM data and the lack of third trimester fetal tissue when gliogenesis should be occurring, the authors do not have the ability to make strong claims about whether gliogenesis or neurogenesis lead to expansion of regions. Consider toning down these conclusions and adding additional support by using glial progenitor genes from time periods that contain these cells (<https://pubmed.ncbi.nlm.nih.gov/37824647/>).

7. Were hyper or hypoallometric genes enriched for GWAS of interindividual differences in cortical surface area or thickness? This analysis seems straightforward and would be an interesting addition to the manuscript.

Reviewer #2

(Remarks to the Author)

The manuscript "Molecular signatures of cortical expansion in the human fetal brain", presents a new resource called μ Brain as a three-dimensional cellular resolution digital atlas combines machine learning enhances histology and gene expression from the BrainSpan, 21 week gestation human dataset. Using serial Nissl-stained and anatomically labelled sections of the

prenatal brain (Ding et al., 2022), the authors performed automated image repair using a generative neural network model before aligning all sections into a common anatomical space, resulting in a 3D volume at 150 μ m voxel resolution. This is an important and creative advance more fully utilizing the fragile and often distorted histology of the atlas and represents a valuable use of the BrainSpan resource. Using the MRI aligned atlas the authors evaluate molecular signatures of preferentially expanded cortical regions during human gestation, quantified in utero using magnetic resonance imaging (MRI). Through this analysis they find that differences in the rates of expansion across cortical areas during gestation respect anatomical and evolutionary boundaries between cortical types. Finally, they identify a set of genes that are upregulated from mid-gestation and highly expressed in rapidly expanding neocortex, which are implicated in genetic disorders with cognitive sequelae.

This is an impressive and valuable atlas derived by using advanced and modern methodology to reconstruct a 3D atlas from imperfect 2D sections, and to use mapped bulk array and partial ISH transcriptomic data to interpret interesting and important findings in developmental neuroscience. While some of these results have been understood through other approaches, the unifying framework that the authors provide is an excellent template for deriving 3D geometric atlases from older and more classical approaches.

The manuscript is very well written and clear demonstrating the authors deep understanding of developmental neuroscience and was a pleasure to read. Care is taken in the methodological and statistical development of μ Brain and I find the work quite rigorous. In summary, the work offers an important approach to data integration in fetal brain neuroscience and I can wholeheartedly support publication of this work.

Reviewer #3

(Remarks to the Author)

This study proposed a method to generate a three-dimensional cellular-resolution digital atlas, μ Brain atlas, using public data of the 21 gestational weeks postmortem human brain sections. The authors detected and repaired artifacts in section images using a Generative Adversarial Network (GAN) and created reconstructed 3D brain volume and cortical surface atlas from 2D serial tissue sections with cortical area labels and gene expression data. Then they found significant correlations between expression level of the set of genes and cortical regional allometric scaling coefficients. This is very interesting study and nice expansion of previous fetal brain – gene studies, such as Vasung et al, 2021. Association between Quantitative MR Markers of Cortical Evolving Organization and Gene Expression during Human Prenatal Brain Development) and Hao et al., 2013. Coupling Diffusion Imaging with Histological and Gene Expression Analysis to Examine the Dynamics of Cortical Areas across the Fetal Period of Human Brain Development (These studies are not referred). However, the research design is very similar to these existing studies. The results and interpretation in this study have limitations due to the inherent limitations of the data. Additionally, the proposed image processing shows limitations in ensuring the reliability of the significant results revealed in this study.

Although this study presented advanced μ Brain atlas, this type of study was previously performed by Hao et al. and Vasung et al. The biggest concern in the results and conclusions is that although cortical area scaling is from 21-38 weeks, the time window for genetic data is too limited. Since temporal variation in gene expression is very high during the fetal stage, it's not sure how to interpret significant spatial correlations between the two datasets strictly. To detect a more advanced and accurate relationship, temporal information on gene expression needs to be included to identify the relationship with cortical expansion. In addition, these two datasets are not from identical subjects. We understand that it's extremely challenging to acquire gene expression and fetal MRI cortical surface data from the same subjects at the same time point. However, we need to admit that the significance and findings of this research are limited due to the limitation of data characteristics.

This study repaired tissue section images with a GAN model and this model requires a lot of training data due to the characteristics of the GAN. This study doesn't seem to include a large dataset for this model. It is doubtful whether the GAN was well trained.

Since the boundaries of cortical area labels were not initially created in 3D, the labels mapped onto the 3D cortical surface of the μ Brain look rough and not very accurate (e.g., motor and sensory areas). Thus, it is doubtful whether labeling on the in-utero fetal MRI cortical surface by surface registration between μ Brain and MRI cortical surfaces is reliable. The cortical surface of the μ Brain and the earliest timepoint fetal template surface are too smooth to have enough cortical folding feature information. This may lead to unreliable surface registration and definition of vertex correspondence between the two cortical surfaces. Moreover, when looking at the 3D shape of the μ Brain, it seems a little different from the typical shape of the 21-22 gestational weeks fetal brain, perhaps because of the distortion. Therefore, it is likely to affect cortical region labeling and spatial correlation between gene expression and regional cortical surface scaling.

There is also concern for the vertex correspondence of individual fetal MRI surfaces. Cortical surface scaling analysis included fetuses from 21 gestational weeks. Due to the lack of cortical folding and highly dynamic changes across age, it is very challenging to define anatomical correspondence across subjects from 21-38 weeks at a vertex level. Since this study performed vertex-wise scaling analysis, it would be essential to show the accuracy and reliability of cortical surface matching.

Version 1:

Reviewer comments:

Reviewer #1

(Remarks to the Author)

The authors have addressed my concerns.

Reviewer #2

(Remarks to the Author)

The authors have responded to the many questions and issues raised by the other reviewers, and I am satisfied with the responses. I can recommend publication at this point.

Reviewer #3

(Remarks to the Author)

The authors tried to address all my comments by clarifying their research process and providing additional information. Some of my concerns have been resolved, but some limitations remain. I have no further comments.

We would like to thank the Reviewers for their thoughtful and constructive comments. Below, we have provided point-by-point responses. In the main manuscript, revisions to the text are highlighted in yellow.

Reviewer #1

This is an interesting manuscript that imputes gene expression data from previously acquired laser capture micro-dissection of the developing human fetal cortex into a 3D atlas, fixing errors in 2D sections through deep learning based imputation. Then, regional gene expression is compared with regional expansion of the brain during development. 116 genes are identified that show a spatial correlation with regional cortical expansion. The genes in fetal development that are spatially correlated with areal expansion are also enriched in genes associated with schizophrenia from GWAS, which is interesting. I think this resource would be widely used for comparison of fetal brain gene expression with fetal and adult MRI, and therefore support the publication. However, I have several concerns about the analyses and interpretations.

1. i. It is surprising in Figs 3f and 4a that genes in the VZ or SVZ (which contain progenitors) have very little spatial correlation with expansion of that cortical region. Most hypotheses about cortical expansion concern progenitor fate decisions, as are discussed here.

R1. While the plots in 3f and 4a do show most significant associations between regional gene expression and rate of (neo)cortical expansion were in the cortical plate, subplate and intermediate zones at 15 and 21PCW, the expression of several genes in proliferative zones were also spatially correlated with cortical expansion. In the text, we note:

"...ZRT_{scaling} genes expressed in the preferentially expanded neocortex were enriched in progenitor cells at 15 PCW (FBXO32, HES6; IPC enrichment = 1.51, p_{hypergeom} = 0.027)..."

Associations shown in **Table S10** identify several genes that were differentially expressed along the allometric scaling gradient in proliferative zones. These included genes associated with stem cell proliferation and differentiation (*ANKRD5, CD99, ANOS1, FAS, PCDH7*, all p_{FDR}<0.01) and extracellular matrix formation and cell adhesion (*PTPRM, EFEMP2, COL9A3, SSPN, FBXO32*, p_{FDR}<0.01), pathways that are crucial to progenitor fate in the germinal zones of the developing brain. Similar associations were observed when focusing on neocortical expansion (**Fig 4a**), most significant associations were localised to the IZ but significant associations were also present in the subventricular and ventricular zones (**Table S10**; 11 associations at p<0.01). Of these, 10/11 were at the earlier 15PCW timepoint. We describe these results in the text as follows:

*"...Consistent with role of prolonged radial glial proliferation in proposed models of cortical expansion, highly expressed ZRT_{neo} genes in areas with a higher rate of expansion over gestation were enriched in radial glia and intermediate progenitors at 15 PCW (p_{hypergeom}=0.045, 0.040 respectively; **Figure S12; Table S11**)..."*

Of the 53 significant ZRT_{neo} associations at 15PCW, 19% were in the proliferative zones, compared to just 1% at 21PCW.

That most associations were observed in post-mitotic tissue may be considered surprising, however we can interpret the data in the following ways:

1. Neurogenesis is largely complete by 21PCW. At this timepoint, we observed few significant associations with cortical expansion in proliferative zones compared to the earlier 15PCW timepoint, with significant enrichment of genes expressed by post-mitotic neurons (**Fig 3g, 4c; Table S10, S11**). The presence of strong areal differentiation in post-mitotic neuron populations does not preclude the prior segregation of progenitor populations at earlier timepoints. It is possible that areal differences in gene expression corresponding to later cortical expansion would be more apparent at earlier stages of neurogenesis in the (sub)ventricular zone, prior to the time window examined here, but we do not have the data at hand to test this hypothesis.
2. Alternatively, previous studies have found that at earlier developmental timepoints, regional gene expression signatures are not as broadly pervasive or area-specific (Bhaduri et al., 2021; Eze et al., 2021). Indeed, prior examination of the LMD microarray used in this study demonstrated far less areal patterning in genes expressed along spatial gradients in germinal compared to post-mitotic layers (**Fig 5** in Miller et al. 2014). As we write in the text (pg. 8):

"...While basic cell types are generally conserved across cortical areas, previous evidence has shown that regional identity is imprinted during cell differentiation, with areal signatures most apparent in post-mitotic cell types..."

As put forward by the Reviewer, evidence suggests that progenitor fate decisions contribute to regional differences in cortical development and growth. Here, we present a snapshot of this process, observing correlations largely between mid-gestation gene expression in post-mitotic layers and subsequent cortical expansion. Our observations are limited by the relatively short time window of the microarray data available to us, lacking temporal sampling over the course of neurogenesis. Our aim is for μ Brain atlas to allow integration of future data to enable further study of cortical development at a whole-brain scale. We have included some comments on these limitations in the Discussion (pg. 14; please see further comments in **R12**, below).

1.ii. In Figure 3g and 4c the authors show an enrichment of hyperallometric ZRT_{scaling} or ZRT_{neo} within progenitors (though less than neurons). This does not seem to match the findings in Figure 3f and 4a because very few genes are spatially correlated with expansion in the zones where progenitors are actually found. How do the authors interpret this finding? Progenitor genes important for regional expansion are correlated with regional expansion not in the progenitors but instead the neurons of the cortical plate where no progenitors exist? These inconsistent findings made the biological conclusions hard to understand.

R2. We would take this opportunity to clarify that in **Fig 3g**, the size and statistical significance of enrichments are illustrated by i) circle size and ii) black outline. This panel shows that hypo- and hyperallometric ZRT_{scaling} genes are only significantly enriched for allocortical (enrichment = 5.25, p=0.002) and neocortical (enrichment = 6.56, p=0.002) neurons, respectively, but not progenitors (p=0.18, 0.37 respectively). While some overlap between gene sets is perhaps expected given that genes are not exclusively expressed by any one cell type, particularly when considering cells with shared lineage, the findings presented in **Fig 3g** and **3f** converge on the same conclusion: *the preferential enrichment in post-mitotic layers of ZRT_{scaling} genes expressed by region-specific neuronal types*. We have added text to clarify this point (pg. 8).

In contrast, **Figure 4c** illustrates cell cluster enrichment of hypo and hyperallometric ZRT_{neo} genes in a UMAP. As the ZRT_{neo} analysis excluded allocortical regions, the previously identified differences (allo- vs neocortical neurons) are

removed and instead we focus on differential enrichments in the neocortex only. In this analysis, we found that at 21PCW hyperallometric ZRT_{neo} genes were enriched for neuronal cell types, whereas hypoallometric area were enriched for progenitors. As noted above, **Fig 4a** shows most significant associations were observed in the post-mitotic IZ. In the Results, we describe this as:

"... ZRT_{neo} genes expressed in the IZ subjacent to preferentially expanded cortical areas were enriched in neuronal populations (enrichment = 2.19, $p_{hypergeom} = 0.00011$) whereas, hypoallometric ZRT_{neo} genes were enriched in proliferative glial cell types (IPC: enrichment=2.96, $p_{hypergeom} < 0.0001$; RG: enrichment=2.36, $p_{hypergeom} < 0.0001$)..."

It is understandable that the results reported in **4a** and **4c** might seem at odds: as discussed in **R1**, several genes expressed by progenitor cells were differentially expressed in regions with differential rates of expansion in the neocortex, and most of those associations were observed in the IZ. In the manuscript, we posit that this reflects proliferative gliogenic processes in the IZ, as follows:

*"... In this case, expression of proliferative cell markers (**Figure 4**) would reflect gliogenesis rather than neurogenesis. To test, we compared ZRT_{neo} genes associated with cortical scaling at 21 PCW in the IZ to region-specific cell type signatures in the mid-fetal brain. Reflecting the proximity to medial allocortex and periallocortical regions, we identified several midline identity genes (*MID1, DMRT5, EMX2*) with high expression in hypoallometric cortex as well as markers of cell proliferation (*HMMR, HAUS6, CENPN, CENPH, PIK3C3*). In support of our hypothesis, we found that hypoallometric ZRT_{neo} genes were specifically enriched in (peri)allocortical glial cell populations (*MDK, SAP30, TMEM98*; astroglia, $p_{hypergeom} = 0.01$; OPC, $p_{hypergeom} = 0.07$)..."*

Expanding on this, of the 30 ZRT_{neo} expressed in hypoallometric IZ, 13 (41%) were expressed by both IPC and RG cell types with 8 additionally expressed by OPCs as well. These included markers of cell division (*CENPH, CENPN, TFD2*) as well as recently identified markers of glial cell lineage (astrocytes: *CRB1, EMX2*; oligodendrocyte: *SMOC1*) (Velmeshev et al., Science, 2023) and genes expressed by newly differentiated oligodendrocytes (*TMEM98, SAP30*). Other genes identified have previously been implicated in progenitor fate decisions and gliogenesis (*ZNF219, EMX2*). Recent spatial transcriptomic studies reveal RG and OPC outside of proliferative zones at 22PCW (Fig S3 in Velmeshev et al., 2023), supporting previous experimental observations of gliogenesis in the IZ during the second half of gestation (Berman et al., 1997; Levers et al. 2001). Together, this evidence consolidates the observations reported in **Fig 4a** and **4c** and lends support to our hypothesis that the enrichment of progenitor genes in the IZ of hypoallometric regions at 21PCW reflects the onset of gliogenesis in these regions. As discussed below (**R8, R12**), we currently lack the data to confirm this hypothesis but evaluating the temporal and spatial regulation of glial fate transition and proliferation during the second half of gestation represents a critical next step for future research.

We have added text the appropriate sections (pg 10,11) to clarify our reasoning and interpretations for the reader.

1.iii. It seems to me that an alternative approach is rather than correlating the level of gene expression within a specific region to expansion of that region, instead correlate the log2FC of the difference between neuronal (CP, SP, IZ) and progenitor containing (VZ, SVZ) for each region with the expansion of that region. Perhaps then the results would be more interpretable.

R3. Please also see comments in **R4** below. The Reviewer suggests testing the association between differential expression across mitotic and post-mitotic zones at mid-gestation and cortical scaling over the third trimester. While this alternative approach removes the ability to interrogate regional differences in gene expression in all tissue zones, it would provide confirmation that ZRT genes expressed in mid-gestation in areas with faster expansion are enriched in post-mitotic cell types (cf.: **Fig 3f, g; Fig 4a, c**).

To test this, we calculated the average differential expression in postmitotic (CP, SP, IZ) vs germinal (SVZ, VZ) tissue for all ZRT genes associated with (neo)cortical scaling ($ZRT_{scaling}$ $n=443$; ZRT_{neo} $n = 116$). We found that, in both genesets, average expression was higher in post-mitotic zones compared to germinal zones for genes positively associated with cortical scaling (hyperallometric genes; $\beta_{zone}=0.68$ 1.11 for $ZRT_{scaling}$ and ZRT_{neo} , respectively; $p<0.001$ both; **Figure R1**). This corresponds with observations in the wider ZRT geneset of increasing gene expression between 15 and 21PCW in postmitotic zones, with corresponding decreases in SVZ and VZ (**Fig 2d & e**, main manuscript)

Figure R1: Differential expression over tissue zones in ZRT genes. Plots show the mean difference in expression of hyper- and hypoallometric $ZRT_{scaling}$ (left) and ZRT_{neo} (right) genes in postmitotic zones (CP, SP, IZ) and germinal zones (SVZ, VZ). Positive values indicate higher expression in postmitotic zones. Horizontal bars indicate median difference with 95% C.I.

We have included this figure in the Supplemental Materials (**Figure S13**) with additional text added to the Results section (pg 8,10).

2. The comparison of genes that are differentially expressed across all zones, time and region with cell types (Figure 2d) conflates these three important variables, and makes the results impossible to interpret. Additionally, the lack of selection for genes upregulated or downregulated again makes the results difficult to interpret. I think it would be more useful to separately compare differentially expressed genes across zones, across regions, and across timepoints with cell types and gene ontologies where there are clear expectations for the outcomes. For example, genes higher expressed in CP > VZ should be neuronal and enriched in neuron populations. I did not find Figure 2e helpful in interpretation because this shows the differences in gene expression within the ZRT genes, but not the function of the genes.

R4. We agree that variations in gene expression and cell type enrichment over zone, region and timepoint each represent important factors in understanding early cortical development and all are interesting avenues for investigation.

We would first like to clarify our intent in identifying ZRT genes for further analysis: this step was to create an initial ‘filter’ to focus our analyses on genes with dynamic (over region, zone and time) changes in expression during our observation window. Our main aim was to identify genes, *in any tissue layer*, with expression correlated to cortical scaling. Using ZRT genes as a starting point ensured that our dataset was enriched for genes with dynamic change – excluding any genes that did not vary significantly over cortical areas or tissue zones (and were therefore not of interest in this study). Indeed, we observed that ZRT genes were 2.8 times more likely to be associated with cortical scaling compared to non-ZRT genes, were 1.4 times more likely to be located near predicted regulatory elements in midgestation, were enriched for genes upregulated in the third trimester and were highly expressed in the postnatal brain compared to non-ZRT genes, supporting a prolonged role in brain growth and development. Thus, the ZRT genes were selected to serve as an initial, enriched geneset to examine the molecular correlates of cortical scaling and brain growth. The analyses summarised in **Figure 2** provide an overall characterisation of the microarray data and the ZRT geneset, illustrating: the main sources of variation in the microarray data (**2a**); up- and down-regulated genes over time in tissue zones and across cell types (**2b**; also **Table S5**); the proportion of all assayed genes differentially expressed over zone, region or time (**2c**); cell type enrichment of ZRT genes (**2d** – now revised to also stratify by tissue zone), and the location of the largest negative and positive age-related changes in ZRT expression by region (**2e**) and zone (**2f**). This is followed by detailed region, zone and time-specific analyses of ZRT_{scaling} and ZRT_{neo} subsets shown in **Figures 3** and **4** and associated supplemental figures.

We value the Reviewer’s suggestion to validate patterns of differential expression against known outcomes to aid interpretation of the microarray data and the initial ZRT geneset. The original analysis of this microarray data (Miller et al., 2014), provided a comprehensive evaluation of differential expression across tissue layers, finding that different layers had robust and unique molecular signatures, with samples grouping by layer after applying a dimension reduction technique to the data. We replicated this analysis in **Figure 2a** (cf. **Fig 2c** in Miller et al.), showing samples grouped into major clusters representing germinal and post-mitotic layers. In **Figure S7**, we illustrated that gene expression was most similar between adjacent mitotic and post-mitotic zones. Miller et al. went on to examine laminar specific gene expression in this dataset, with corresponding gene ontologies illustrating the expected developmental hierarchy across layers – from cell division and gliogenesis in the proliferative zones, to neuronal markers and synaptic transmission in post-mitotic layers (**Fig 2d**, Miller et al.). At the Reviewer’s suggestion, we have repeated our cell-type specific enrichment analysis originally shown in **Figure 2d** to complement this prior work, focusing on ZRT genes that are up-regulated in each tissue zone compared to others. The results of this analysis are now shown in a revised **Figure 2d** and **Figure S10** (and below in **Figure R2** for reference). We found cell-type enrichments followed expected patterns, with proliferative cell types enriched in germinal zones and neuronal cell types enriched in post-mitotic zones demonstrating that the ZRT geneset captures known developmental trajectories of cell type maturation and migration across cortical tissue layers.

Figure R2: Cell-type enrichment of ZRT genes differentially expressed in the cortical plate and the ventricular zone. Significant enrichments of each cell type are indicated by black outlines. UMAPs shown cell cluster enrichment for each set of ZRT genes. Results for each zone are shown in Figure S10.

We have added some text to the appropriate section to include these results, point interested readers to previous characterisations of the data by Miller et al and to better contextualise our observations and interpretations (pg 5,6).

3. In Figure 1g it would be helpful to have a more zoomed in representation of the ISH reconstructions. It is surprising that EOMES is not consistently expressed across the SVZ of the dorsal cortex and only found in the ventral cortex.

R5. See below for a zoomed in representation of the ISH maps (**Fig R3**) and raw EOMES ISH data from the BrainSpan atlas (**Fig R4**) illustrating EOMES distribution in the 21PCW brain. We will make the partial ISH reconstructions available on our Zenodo repository to allow other researchers to explore the volumes at high-resolution. We would like to confirm that in more caudal sections, *EOMES* is expressed in both dorsal and ventral cortex. We have amended **Figure 1** to incorporate a more representative section for this stain.

Figure R3: Serial sections through the partial reconstruction of EOMES ISH data. Higher expression is indicated by brighter colours. Sections are arranged in order from anterior (left) to posterior (right).

Figure R4: Raw ISH data with corresponding Nissl sections from the BrainSpan atlas. Two posterior sections are shown with EOMES ISH staining.

4. ZRT genes are labeled as Zone-Region-Tissue in the text, but I believe it should be Zone-Region-Timepoint to match Figure 2c.

R6. Thanks for pointing this out. We have corrected the text where needed.

5. It would especially be interesting if ZRT_{neo} overlapped with known signaling pathways involved in areal identity in the cortex or known areal markers from scRNA-seq (<https://www.nature.com/articles/s41586-021-03910-8>)

R7. We agree with the Reviewer that this is an interesting possibility. To test this, we quantified the enrichment of ZRT_{neo} genes in the list of cell-specific area markers identified by Bhaduri et al (Table S8 in the referenced paper). We found a significant overlap between ZRT_{neo} genes and neuron-specific areal markers (34/116 genes, enrichment=1.34, $p_{\text{hypergeom}}=0.037$) and, to a lesser extent, intermediate progenitor areal markers (18/116, $p_{\text{hypergeom}}=0.045$) but not radial glia areal markers ($p_{\text{hypergeom}} = 0.257$). The same pattern was seen in ZRT_{scaling} genes ($p_{\text{hypergeom}} = 0.034, 0.047, 0.272$ for neuron, IPC and RG, respectively). We have included these results in the revised manuscript (pg 10).

6. With the lack of cell type specific resolution of the LCM data and the lack of third trimester fetal tissue when gliogenesis should be occurring, the authors do not have the ability to make strong claims about whether gliogenesis or neurogenesis lead to expansion of regions. Consider toning down these conclusions and adding additional support by using glial progenitor genes from time periods that contain these cells(<https://pubmed.ncbi.nlm.nih.gov/37824647/>).

R8. We agree that, given the available data, we are unable to determine whether cortical expansion is founded upon regional differences in neurogenesis or gliogenesis. In our Discussion we state:

"...While our data suggest rapid areal expansion is preceded by an extended neurogenic period, we lack the data to confirm a similarly extended period of gliogenesis. In humans, neurogenesis precedes cortical folding with the subsequent gliogenic period more closely aligned to the timing of cortical expansion. An extended neurogenic period coupled with a longer migration time due to the expanding volume of the brain may necessitate an extended gliogenic period to populate the expanding neuropil. Evaluating the temporal and spatial regulation of glial fate transition and proliferation in the oSVZ during the second half of gestation represents a critical next step in understanding this process..."

We have also added additional comments on the limitations of the available data (see also **R11** and **R12** below), e.g.:

"...In this study, our observations are limited by available microarray data to a short gestational window, precluding a fine-grained examination of concurrent changes to gene expression and cortical folding over the full third trimester..."

And have added qualifying statements where appropriate in the Results and Discussion sections to ensure clarity around our interpretation of the data (pg 10, 12).

In addition, we thank the Reviewer for highlighting this excellent paper, we have added it to the Discussion (pg. 13). We have also used the data included to investigate the presence of glial lineage genes in ZRT_{neo} genes, as suggested (pg 11; see also **R2**). As noted elsewhere (**R12**), the lack of available data with both detailed spatial and temporal sampling precludes a spatial analysis of this type across the full gestation period. We hope that future studies can be integrated into the μ Brain coordinate framework to facilitate further comparisons and research into this area.

7. Were hyper or hypoallometric genes enriched for GWAS of interindividual differences in cortical surface area or thickness? This analysis seems straightforward and would be an interesting addition to the manuscript.

R9. We thank the Reviewer for this excellent suggestion. We have performed this additional analysis using the available bilateral cortical thickness and area GWAS summary statistics (using the Desikan-Killiany atlas) available through the ENIGMA consortium (<https://enigma.ini.usc.edu/research/download-enigma-gwas-results/>; Grasby et al., 2020, Science). We used MAGMA (de Leeuw et al., 2015, PLOS Computational Biology) to perform gene-set enrichment analyses using the hyper- and hypo-allometric scaling gene sets. Results are shown below, and demonstrate no enrichment for regional cortical thickness and sparse specified enrichment for cortical surface area (**Figure R5**). We have included these results in the main manuscript (pg 13) and as an additional supplemental figure (**Figure S19**).

Figure R5. Bilateral cortical thickness (CT) and surface area (SA) GWAS enrichment results of hyper- and hypo-allometric scaling gene sets. Beta coefficients for gene-set enrichment analyses performed with MAGMA are plotted on cortical surfaces (left hemisphere shown just for visualization). Black outlines denote significant enrichment ($p < 0.05$, uncorrected).

Reviewer #2

The manuscript “Molecular signatures of cortical expansion in the human fetal brain” , presents a new resource called μ Brain as a three-dimensional cellular resolution digital atlas combines machine learning enhances histology and gene expression from the BrainSpan, 21 week gestation human dataset. Using serial Nissl-stained and anatomically labelled sections of the prenatal brain (Ding et al., 2022), the authors performed automated image repair using a generative neural network model before aligning all sections into a common anatomical space, resulting in a 3D volume at $150\mu\text{m}$ voxel resolution. This is an important and creative advance more fully utilizing the fragile and often distorted histology of the atlas and represents a valuable use of the BrainSpan resource. Using the MRI aligned atlas the authors evaluate molecular signatures of preferentially expanded cortical regions during human gestation, quantified in utero using magnetic resonance imaging (MRI). Through this analysis they find that differences in the rates of expansion across cortical areas during gestation respect anatomical and evolutionary boundaries between cortical types. Finally, they identify a set of genes that are upregulated from mid-gestation and highly expressed in rapidly expanding neocortex, which are implicated in genetic disorders with cognitive sequelae.

This is an impressive and valuable atlas derived by using advanced and modern methodology to reconstruct a 3D atlas from imperfect 2D sections, and to use mapped bulk array and partial ISH transcriptomic data to interpret interesting and important findings in developmental neuroscience. While some of these results have been understood through other approaches, the unifying framework that the authors provide is an excellent template for deriving 3D geometric atlases from older and more classical approaches.

The manuscript is very well written and clear demonstrating the authors deep understanding of developmental neuroscience and was a pleasure to read. Care is taken in the methodological and statistical development of μ Brain and I find the work quite rigorous. In summary, the work offers an important approach to data integration in fetal brain neuroscience and I can wholeheartedly support publication of this work.

R10. We thank the Reviewer for their positive comments.

Reviewer #3

This study proposed a method to generate a three-dimensional cellular-resolution digital atlas, μ Brain atlas, using public data of the 21 gestational weeks postmortem human brain sections. The authors detected and repaired artifacts in section images using a Generative Adversarial Network (GAN) and created reconstructed 3D brain volume and cortical surface atlas from 2D serial tissue sections with cortical area labels and gene expression data. Then they found significant correlations between expression level of the set of genes and cortical regional allometric scaling coefficients.

This is very interesting study and nice expansion of previous fetal brain – gene studies, such as Vasung et al, 2021. Association between Quantitative MR Markers of Cortical Evolving Organization and Gene Expression during Human Prenatal Brain Development) and Hao et al., 2013. Coupling Diffusion Imaging with Histological and Gene Expression Analysis to Examine the Dynamics of Cortical Areas across the Fetal Period of Human Brain Development (These studies are not referred). However, the research design is very similar to these existing studies. The results and interpretation in this study have limitations due to the inherent limitations of the data. Additionally, the proposed image processing shows limitations in ensuring the reliability of the significant results revealed in this study.

R11. We agree that our work complements previous studies of the fetal brain such as those by Huang et al and Vasung et al. We regret the omission of these works from our submission and have now amended the text to appropriately cite these important studies (pg. 7 & 14). We would also like to take this opportunity to highlight the novel aspects of our approach:

1. *In vivo characterisation of the fetal brain using a large fetal MRI database.* Unlike previous similar studies that have examined the prenatal brain using *postmortem* MRI (Huang et al, 2013) or relatively small *in utero* samples (Vasung et al. 2021, n=25 between 20-33 weeks gestation), we take advantage of the recently acquired dHCP fetal dataset comprising n=195 scans from 21 to 38 weeks gestation. This provides a large and robust sample to estimate cortical expansion over the third trimester at high-resolution.
2. *A comprehensive spatial survey of prenatal cortical gene expression.* Previous studies (Vasung et al. 2021; Huang et al 2013; Pecheva et al., 2020), including our own (Ball et al., 2020), have relied upon the BrainSpan/PsychENCODE bulk tissue RNA-seq data (Kang et al., 2011; Li et al., 2018) to examine prenatal imaging-transcriptome associations. These data are sampled at 11 regions across the neocortex, inclusive of cortical plate, subplate and proliferative zones. Through construction of the μ Brain atlas, we are now able to perform imaging-transcriptomic analyses of microarray expression across 29 cortical regions and five transient tissue zones, allowing a far more comprehensive examination of these relationships than previously possible.
3. *An open access resource for researchers.* By providing the μ Brain atlas and underlying data for others to download, we generate a resource to facilitate future research in this area, maximising value from these rare data sources.

As noted by the Reviewer, our work is not without limitations, many that are inherent to the data at hand: regional microarray data are only available at 15 and 21 PCW; the fetal MRI data covers only the second half of gestation, and the *postmortem* microarray data and *in vivo* neuroimaging are from different subjects.

However, as we write in the Discussion:

"...spatially-embedded gene expression atlases of the adult human and mouse have proven exceptionally powerful in recent years, bridging resolution gaps to common neuroimaging modalities and providing insight into the molecular correlates of structural and functional neuroanatomy, brain development, disease and disorder. In such studies, comparisons with in vivo neuroanatomy can only be fully realised through three-dimensional localisation of tissue samples within a common coordinate space. To date, a limitation of this approach has been either the sampling of a narrow age range outside of key developmental periods or, in developmental datasets, a lack of 3D spatial information and relatively coarse anatomical sampling...."

In other words, there is significant power in gene expression atlases to further our understanding of brain organisation, development and disorder despite the known limitations. While the μ Brain atlas addresses some of this in the fetal brain, namely the spatial embedding of genetic data and improved anatomical sampling, we acknowledge that several limitations remain and have added comments in the Discussion to make clear the limits of our interpretations given the data at hand (pg. 14). We and others look forward to the availability of future data resources with a continued focus to tackle these remaining limitations.

Although this study presented advanced μ Brain atlas, this type of study was previously performed by Hao et al. and Vasung et al. The biggest concern in the results and conclusions is that although cortical area scaling is from 21-38 weeks, the time window for genetic data is too limited. Since temporal variation in gene expression is very high during the fetal stage, it's not sure how to interpret significant spatial correlations between the two datasets strictly. To detect a more advanced and accurate relationship, temporal information on gene expression needs to be included to identify the relationship with cortical expansion. In addition, these two datasets are not from identical subjects. We understand that it's extremely challenging to acquire gene expression and fetal MRI cortical surface data from the same subjects at the same time point. However, we need to admit that the significance and findings of this research are limited due to the limitation of data characteristics.

R12. As noted above, we agree that our observations are limited by the relatively short time window of the microarray data available to us. One of the main aims of this study was to examine how regional patterns of gene expression at *mid-gestation* are related to *subsequent growth* of the cortical surface in the third trimester. Examination of the *concurrent* changes in regional gene expression and cortical folding over gestation, as suggested by the Reviewer, represents a related but independent research question.

Unfortunately, despite a growing catalogue of comprehensive single-cell RNA-seq studies, including those that span the whole of gestation (e.g.: Herring et al., 2022; Velmeshev et al., 2023), there are no datasets of fetal gene expression that encompass 20-40 weeks over multiple cortical regions at a spatial resolution comparable to the LMD microarray data used in μ Brain. Previously, using the BrainSpan RNA-seq data, we have examined the associations between variation in gene expression between 12 and 37 weeks and cortical morphology in neonates (Ball et al., PLoS Biology, 2020). Others,

as the Reviewer notes, have used this data to examine cortical growth *in utero* (Vasung et al., 2021). However, this data is limited to just 11 cortical regions-of-interest, each encompassing tissue from the full cortical anlage (cortical plate, subplate, proliferative zones). In contrast, the LMD microarray data used in the present study has exquisite spatial resolution, with samples from multiple cortical regions and tissue zones, but lacks temporal sampling. We await a dataset that offers both spatial and temporal resolution during this time period, recognising that this is extremely challenging data to acquire. We hope that future studies can be integrated into the μ Brain coordinate framework by matching anatomical sampling locations, for example, to facilitate further comparisons. In the meantime, we have made clear the limitations of these data in the Discussion (pg. 14).

This study repaired tissue section images with a GAN model and this model requires a lot of training data due to the characteristics of the GAN. This study doesn't seem to include a large dataset for this model. It is doubtful whether the GAN was well trained.

R13. As with many image-based generative models, the Reviewer is correct in the assertion that GAN models can often require large datasets to train. However, other parameters can also determine the size of dataset needed, for example: the quality of the training data, the size and complexity of the images and the variance across image classes.

Here, we trained the *pix2pix* GAN using n=1000 curated image-pairs (now clarified on pg.2). Each image represented a 256 x 256 pixel patch drawn from one of the histological sections and its corresponding tissue label image at random. Each pair was inspected to ensure that there was good alignment between tissue labels and Nissl-stained sections and no image artefacts. In addition, we implemented data augmentation transformations (jitter, cropping and flipping) to each image pair to increase training data size and diversity. The performance of the GAN model in image patches drawn from sections unseen during training is illustrated in **Fig S2**, **Fig 1c** and **Fig 1d**. Close-up sections from **Fig 1c** are provided below for reference (**Figure R6**). The GAN model was clearly able to produce image patches matched in hue, saturation and structure to the original data (**Fig S2; Figure R6, right**).

We can confirm that in the original *pix2pix* paper (Isola et al. 2016), the model was successfully trained on a range of image datasets of similar size to ours including: CityScapes (n=2975 samples), CMP Façade (n=400 samples) and Google Maps (n=1096 samples). Therefore, we are confident that our GAN implementation was sufficiently well trained to produce synthetic Nissl-stained image patches with realistic colour and contrast to be used for reconstruction.

Figure R6: Model performance in a validation section (unseen during model training). Left: 256 x 256 labelled image patches were passed to the trained model to predict corresponding Nissl-stained patches. Left image shows the ground truth, middle illustrates a checkerboard occlusion, right shows ground truth alongside model predictions in occluded patches. Right. Zoomed in images of the ground truth and model predictions for patches highlighted in red and yellow. Note similarities in colour, contrast, structure and texture and how the artefact in the red patch has been repaired in the model prediction, illustrating a well-trained GAN model.

Since the boundaries of cortical area labels were not initially created in 3D, the labels mapped onto the 3D cortical surface of the μ Brain look rough and not very accurate (e.g., motor and sensory areas). Thus, it is doubtful whether labeling on the in-utero fetal MRI cortical surface by surface registration between μ Brain and MRI cortical surfaces is reliable. The cortical surface of the μ Brain and the earliest timepoint fetal template surface are too smooth to have enough cortical folding feature information. This may lead to unreliable surface registration and definition of vertex correspondence between the two cortical surfaces. Moreover, when looking at the 3D shape of the μ Brain, it seems a little different from the typical shape of the 21-22 gestational weeks fetal brain, perhaps because of the distortion. Therefore, it is likely to affect cortical region labeling and spatial correlation between gene expression and regional cortical surface scaling.

R14. We agree with the Reviewer that the boundaries of the μ Brain cortical labelling, on its native surface (**Fig 1f**), do not necessarily reflect the expected position of neuroanatomical borders of the prenatal brain. This is due to a) the anatomical deformations of the cerebral volume from fixation and dissection b) the interleaved sampling strategy of the BrainSpan atlas leading to an uneven distribution of sections through the whole hemisphere (Ding et al., 2022) and c) the subsequent exclusion of sections at the tissue or image processing stages. Our strategy to mitigate some of these effects included a) repair of large tissue warps and tears using the GAN model b) the use of a fetal brain template to constrain volumetric registration during reconstruction and c) creating a smooth anatomical representation by oversampling and augmentation of the initial aligned volume. Details are included in the Supplemental Methods. While these strategies were able to produce a high-resolution 3D image with size and volume matched to expected fetal brain norms (Papageorgiou et al., 2018; Scott et al., 2011), in practice the volume requires subsequent nonlinear alignment to a target for accurate delineation of cortical boundaries. For this, we employed a two-stage strategy outlined on pg 19 of the manuscript and illustrated in **Figure S18**.

In brief, we used MSM to perform an initial spherical registration between μ Brain and the 22PCW dHCP surface based on alignment of sulcal depth. As noted by the Reviewer, due to the lack of cortical folding this alignment is driven largely by the large-scale anatomical features of each surface – the frontal and occipital poles, the operculum, etc. To improve cortical alignment, we generated a set of cortical labels, defined by major sulcal anatomy on the 36 week template surface and transferred down to each timepoint, with corresponding labels on the μ Brain surface. Using these 11 cortical ROI, we calculated a second registration, initialised using the previously calculated sulcal alignment and driven by alignment of cortical ROIs across surfaces. This approach leverages anatomical labels (defined based on cytoarchitecture or using older fetal anatomy in μ Brain and dHCP atlases, respectively), to inform cortical alignment in the absence of geometric features. A similar approach has proven successful accommodating large deformations across primate species (Eichert et al., eLife, 2020). The alignment of cortical ROIs before and after the second registration are shown in **Figure S18**. We have added a comment to clarify this point in the Discussion (pg 14).

The cortical parcels shown in **Fig 1f** are the projections of the original tissue annotations projected onto the μ Brain cortical surface. In contrast, the cortical parcellations in **Fig 3b** show the atlas at five timepoints after alignment to the dHCP cortical template. Anatomical correspondence is clear across timepoints, with parcel boundaries aligning with major neuroanatomical borders. An individual parcellation at 36w is shown below (**Fig R7**). After the two-stage cortical registration, the μ Brain labels are well positioned on the individual's surface and aligned to major sulci. As discussed below (see **R15**), we aim for future research to improve cortical surface registration and parcellation, particularly in a longitudinal setting.

Figure R7: Cortical parcellation of an individual surface at 36 weeks gestation. Major neuroanatomical boundaries are illustrated. Although cortical labels were defined based on histological data at 21 PCW, the two-stage registration allows for alignment of labels with cortical folds at later ages. For parcellations at earlier timepoints, see **Fig R9** below.

Regarding the shape of the 3D volume, **Fig 1** shows the μ Brain volume in its native space after initial reconstruction. While this volume reflects the size of the fetal brain and is matched in length and volume to expected norms, it appears narrower than expected when compared to fetal MRI of equivalent age. As stated above, this likely reflects various nonlinear distortions of the postmortem tissue. As with the cortical surface, we feel it is important to visualise the reconstructed volume after processing, without further registration steps, to illustrate the endpoint of the reconstruction process. That said, we can perform a secondary nonlinear alignment to an appropriate volumetric template to create a volume with the expected shape – albeit with lower resolution (see **Figure R8**). We are able to make this version available for others to use on our public repository.

Figure R7: The μ Brain volume following nonlinear registration to the 22 week dHCP fetal volumetric template.

There is also concern for the vertex correspondence of individual fetal MRI surfaces. Cortical surface scaling analysis included fetuses from 21 gestational weeks. Due to the lack of cortical folding and highly dynamic changes across age, it is very challenging to define anatomical correspondence across subjects from 21-38 weeks at a vertex level. Since this study performed vertex-wise scaling analysis, it would be essential to show the accuracy and reliability of cortical surface matching.

R15. We agree that cortical surface alignment of fetal data is a challenging task, partly due to the rapidly changing complexity of the cortical surface during the third trimester. Indeed, in our Discussion we state:

“...A key area for future research in this field is the development and validation of improved methods to align early MRI to common template spaces. The geometry of the fetal cortex is smooth, making alignment of cortical morphometry an ill-posed problem. Newer, anatomically-constrained registration techniques and larger longitudinal cohorts with multiple scans during mid- to late-trimester will enable more precise estimates of cortical expansion in the future...”

Considering this, we took several steps to ensure accuracy of our cortical registrations and subsequent analysis:

1. We aligned individual cortical surfaces to their nearest timepoint in the spatiotemporal dHCP fetal atlas. This means that no registrations are required between e.g.: 21 to 38 weeks, all registrations are performed using an age-appropriate template with similar degree of cortical folding as a target.
2. After resampling, all cortical surfaces and metrics (e.g. curvature) were visually checked to ensure quality of alignment.
3. We applied a vertexwise outlier detection strategy (see **Figure S20**) to identify any vertices where surface area was much larger than expected for a given cortical region at a given age. Outlier vertices were identified using a sliding window over age (outliers >2.5 S.D. from the mean within a given window, maximum window size=25 scans, sorted by age).
4. To mitigate residual misalignment, vertexwise estimates of area were smoothed with a Gaussian kernel (FWHM = 10mm) prior to analysis
5. We compared groupwise estimates of cortical expansion to that derived from a single subject, scanned three time during gestation (**Figure S12**).

To illustrate accuracy of the resulting registrations, a selection of individual surfaces, parcellated with the μ Brain cortical atlas are shown below (**Figure R9**). The zoomed images illustrate labelling of the M1/S1 boundary, note the labels respect the position and folding of the central sulcus across individual at different ages.

We would like to highlight that while we calculated vertexwise scaling models, the main analysis compared cortical scaling within cortical regions. By averaging scaling data within cortical parcels the effects of any residual vertex-level misalignment would be further mitigated. As noted in the Discussion, future research will aim to improve cortical surface registration, particularly in a longitudinal setting. Through our freely available online data repository, we will have the capacity to publish updated, versioned μ Brain atlas parcellations as these advances take place.

Figure R9: Individual cortical parcellations using the μ Brain atlas. Individual mid-thickness surfaces are shown for 10 individuals with age spanning the third trimester, with topology resampled onto the dHCP 32k template surface. For each surface, μ Brain cortical labels are shown for medial (left) and lateral (right) surfaces. The zoomed inset illustrates cortical labelling of M1 and S1 cortex on each surface.